# The Gain and Loss of Cryptochrome/Photolyase Family Members during Evolution

**DOI:** 10.3390/genes13091613

**Published:** 2022-09-08

**Authors:** Peter Deppisch, Charlotte Helfrich-Förster, Pingkalai R. Senthilan

**Affiliations:** Neurobiology & Genetics, Theodor-Boveri Institute, Biocenter, Julius-Maximilians-University Würzburg, 97074 Wurzburg, Germany

**Keywords:** cryptochrome, photolyase, cryptochrome/photolyase family, CPF, CRY evolution, circadian clock, DNA-repair

## Abstract

The cryptochrome/photolyase (CRY/PL) family represents an ancient group of proteins fulfilling two fundamental functions. While photolyases repair UV-induced DNA damages, cryptochromes mainly influence the circadian clock. In this study, we took advantage of the large number of already sequenced and annotated genes available in databases and systematically searched for the protein sequences of CRY/PL family members in all taxonomic groups primarily focusing on metazoans and limiting the number of species per taxonomic order to five. Using BLASTP searches and subsequent phylogenetic tree and motif analyses, we identified five distinct photolyases (CPDI, CPDII, CPDIII, 6-4 photolyase, and the plant photolyase PPL) and six cryptochrome subfamilies (DASH-CRY, mammalian-type MCRY, Drosophila-type DCRY, cnidarian-specific ACRY, plant-specific PCRY, and the putative magnetoreceptor CRY4. Manually assigning the CRY/PL subfamilies to the species studied, we have noted that over evolutionary history, an initial increase of various CRY/PL subfamilies was followed by a decrease and specialization. Thus, in more primitive organisms (e.g., bacteria, archaea, simple eukaryotes, and in basal metazoans), we find relatively few CRY/PL members. As species become more evolved (e.g., cnidarians, mollusks, echinoderms, etc.), the CRY/PL repertoire also increases, whereas it appears to decrease again in more recent organisms (humans, fruit flies, etc.). Moreover, our study indicates that all cryptochromes, although largely active in the circadian clock, arose independently from different photolyases, explaining their different modes of action.

## 1. Introduction

### 1.1. The Cryptochrome/Photolyase Family (CRY/PL Family)

Depending on its wavelength and intensity, light can be both vital and life-threatening. To have fast, sensitive, yet broad light perception, many organisms possess a variety of light-sensitive molecules including opsins and cryptochromes. Cryptochromes are highly conserved flavoproteins belonging to the cryptochrome/photolyase family (CRY/PL family or CPF) [1]. The CRY/PL family consists of several UV-A/blue light-sensitive proteins with common structures and chromophores divided into photolyases (PLs) and cryptochromes (CRYs) [2]. Both are similar in sequence and structure but have different physiological functions (Figure 1). Photolyases are evolutionarily ancient, light-activated enzymes that repair DNA damage caused by UV radiation. Most cryptochromes have lost the DNA repair function and gained new roles as light receptors, transcriptional regulators, magnetoreceptors and particularly as key players in circadian clocks. Some cryptochromes have deepened their role even to the point of losing their sensitivity to light. Due to the simultaneous discovery of several CRY/PLs in different organisms and due to different isoforms and duplications, the CRY/PL nomenclature is inconsistent. We briefly explain each of the subfamilies below.

### 1.2. 6-4 & CPD Photolyases (6-4 PL & CPD PL)

Photolyases recognize and repair UV-light induced DNA damages with blue light via cyclic electron transfer, breaking bonds and restoring DNA integrity in the process (Figure 1A,B) [2,3,4,5]. While 6-4 photolyases repair pyrimidine-pyrimidone (6-4) photoproducts [6], CPD photolyases restore cyclobutane-pyrimidine dimers [7]. The CPD photolyases are further divided into three subclasses (I, II, and III), with subclasses I and III being quite similar and II forming a distinct group [8,9,10].

### 1.3. Plant-Type CRY (PCRY)

*Arabidopsis thaliana* PCRY is the first identified cryptochrome [11]. It works as UV-A/blue-light receptor that forms dimers in the dark and splits into monomers upon light activation (Figure 1C). The activated PCRY monomers transmit blue-light signals to downstream signaling pathways, e.g., to that of the E3 ubiquitin ligases COP1 (CONSTITUTIVE PHOTOMORPHOGENIC1) and SPA (SUPPRESSOR OF PHYA-105). In the dark, COP1 and SPA form active complexes that promote the degradation of the transcription factors HY5 (ELONGATED HYPOCOTYL5) and CO (CONSTANS). Light-activated PCRYs inactivate the COP1/SPA complex, allowing HY5 and CO to accumulate in the nucleus and activate transcription of downstream genes. PCRYs can also directly activate CIB, a basic helix-loop-helix (bHLH) transcription factor [12].

### 1.4. Animal Cryptochromes: Drosophila-Type & Mammalian-Type CRY (DCRY & MCRY)

The *Drosophila*-type CRYs (DCRYs, also called CRY1s or type I CRYs) and the mammalian-type CRYs (MCRYs, also called CRY2s, vertebrate CRYs or type II CRYs) are key molecules of the circadian clock, though affecting it differently. While MCRYs have lost their capability to perceive light, DCRYs remain light-sensitive (Figure 1D). Absorption of light by the flavin cofactor FAD leads to a conformational DCRY change [13] that enables interaction with TIMELESS (TIM) driving its degradation. TIM, a key protein in the *Drosophila* clock, interacts with PERIOD (PER) and as heterodimers, both inhibit their own transcription, resulting in a transcription and translation feedback cycle of about 24 h [14,15,16]. The light-insensitive MCRYs were first discovered as 6-4 PL homologs in human cells [17] and subsequently characterized in the mouse *Mus musculus* for the first time [18]. Here, they took over the function of *Drosophila* TIM and act directly as transcriptional repressors, inhibiting their own transcription (Figure 1E). The light-insensitive MCRYs, first discovered in the mouse *M. musculus* [18] overtook the function of *Drosophila* TIM and work as transcriptional repressors, inhibiting their own transcription (Figure 1E). Due to their high similarity, especially in terms of protein sequences, DCRYs and MCRYs are often grouped together as animal CRYs [19,20,21]. 

### 1.5. DASH-CRY

DASH-CRYs were first discovered by [22] in the cyanobacterium *Synechocystis* sp. They have their name due to the sequence similarities between the cryptochromes of *D. melanogaster*, *A. thaliana*, *Homo sapiens*, and *Synechocystis sp*. (Figure 1F). DASH-CRYs are evolutionary related to 6-4 PLs and animal CRYs (MCRYs & DCRYs) [22,23]. Initial in vitro experiments have demonstrated the ability of DASH-CRYs to repair CPDs in single-stranded but not in double-stranded DNA [24]. In the fungus *Phycomyces blakesleeanus*, DASH-CRY was shown to repair CPD lesions in dsDNA as well [25,26]. DASH-CRYs seem also to be involved in the circadian clock of the mold *Neurospora crassa* that possesses a CRY-dependent oscillator besides the main oscillator that depends on FREQUENCY and WHITE COLLAR 1 [27].

### 1.6. Other CRY/PL Family Members (PCRY-Like, CRY4, and PPL)

In addition to the larger groups, further CRY/PL family members occur that are less explored or are found only in smaller subpopulations. PCRY-like (plant-CRY-like) was first described in diatoms and subsequently other orthologs were discovered in aquatic metazoans [28,29,30]. Even though PCRY-like proteins cluster with PCRY sequences, the definite function of PCRY-like proteins has not yet been unraveled. In the diatom *Phaeodactylum tricornutum* and the zebrafish *Danio rerio* a role of PCRY-like in circadian rhythm regulation is quite likely, as its expression is cycling over the day [29]. CRY4 (CRYPTOCHROME 4), first identified in zebrafish [31] and frequently found in migratory animals, seems to be involved in magnetoreception [32,33]. The plant photolyase PPL (also known as PHR2) of *Chlamydomonas reinhardtii* has been shown to repair CPD damages in both chloroplast and in nuclear DNA [34]. In addition, species-specific cryptochromes are also present in more ancient metazoans such as porifera and cnidaria.

### 1.7. Aim of This Project

Even closely related species differ in number and types of CRY/PL members. While *D. melanogaster* (Diptera) has one DCRY and two photolyases: 6-4 and CPDII PLs [35], the honeybee *Apis mellifera* (Hymenoptera) possesses only MCRY and CPDII [36,37]. Thus, the CRY/PL complement of honeybees is more akin to that of mammals rather than to closely related flies. In this study, we want to understand the evolutionary relationship between CRY/PL composition and mode of life of different species. We address the following questions: (1) Which were the first members of the CRY/PL family? (2) Which members have newly evolved and which members were lost during evolution? (3) What circumstances favor the presence of certain CRY/PL family members? Several studies have already examined the phylogenetic relationships between various CRY/PL members [19,20,38]. Other studies have additionally investigated the specific roles of CRY/PL members in particular species [29,30,39,40]. Yet, these studies either lack certain recently discovered members of the CRY/PL-family or worked with a restricted number of species, mainly model organisms. In our study, we have utilized the large number of already sequenced and annotated genes and systematically screened them for their CRY/PL family members. Our primary focus is on metazoans, as they and their CRY/PL family members are widely known and studied.

## 2. Materials and Methods

### 2.1. Tree of Life

For the construction of a representative phylogenetic tree, we first searched for already sequenced and annotated organisms belonging to different taxa. For this purpose, we chose the NCBI taxonomy tool (www.ncbi.nlm.nih.gov/taxonomy, accessed between 1 September 2021 and 15 September 2021) and the Tree of Life Web Project (tolweb.org/tree, accessed between 1 September 2021 and 15 September 2021) and assigned the found species according to their scientific classifications. In this work, we focused mainly on the Metazoa taking the other kingdoms and domains primarily as references. We have listed all organisms studied and their taxonomic assignment on Appendix A. The exact number of organisms examined per taxonomic level is further indcated in the respective results.

### 2.2. BLASTP

We found putative homologs of CRY/PL family members using the BLASTP program from NCBI (blast.ncbi.nlm.nih.gov/Blast.cgi?PAGE=Proteins, accessed between 15 September 2021 and 15 February 2022). For this, we used the amino acid sequences of *D. melanogaster* CRYPTOCHROME (NP_732407) and *D. melanogaster* PHOTOREPAIR (NP_523653) as reference sequences. For BLASTP search, we chose the non-redundant protein sequences (nr) as the search set and BLASTP (protein-protein BLAST) as the algorithm with its default parameters (BLOSUM62, Gap Costs 11/1, Conditional compositional score matrix adjustment). To obtain sequences of more than just closely related organisms, we performed several searches in succession, each time restricting the organisms to be searched to specific taxonomies (kingdom, phylum, and class). In order not to miss any sequences, especially for extensively sequenced phyla and orders, we set the maximum results per search (Max target sequences to 1000 sequences. For our analysis, we considered all putative CRY/PL sequences sequenced and annotated until February 2022. These sequences had different origins; some sequences originated from complete or partial genomes, some from transcriptomes, and some from studies that examined CRY/PLs in specific organisms and sequenced and annotated them. We collected the analysis results in Excel and transferred the amino acid sequences into the Geneious Prime software 2021.2.2. Here, we only took the sequences that had an e-value less than e-15. To maintain a reasonably balanced ratio, we reduced the number of orders and families with many sequenced species. During this reduction, we first tried to keep known and already studied species. The other species in an order and family were selected purely randomly, according to their order in the BLASTP search. At the end of this process, we obtained more than 4000 protein sequences from 639 organisms, which also contained erroneous and multiple annotations that we purified in the next step (2.3). The final 2249 purified sequences and their accession numbers are provided in Appendix A.

### 2.3. Phylogenetic Tree

The phylogenetic tree was created using the Geneious Prime software 2021.2.2. We applied the Alignment type: Global alignment with the Cost Matrix PAM 100 and the gap costs 10/1. We selected Jukes-Cantor as Genetic Distance Model and Neighbor-Joining as Tree Build Method. To avoid protein isoforms, gene duplications, and multiple annotations of the same proteins, the analysis was repeated several times with removal of duplicate sequences of the same gene. Simultaneously, protein sequences that turned out to be outliers in the phylogenetic tree were verified by reciprocal-BLAST to *Drosophila melanogaster* to confirm that they are indeed CRY/PL family members. We also checked the accuracy of sequences that arranged very well in particular clusters by randomly selecting and reciprocal-BLAST search some of them. The original phylogenetic trees contained more than 4000 protein sequences, including erroneous annotations as well as isoforms, which we subsequently cleaned up manually. To do this, we first examined the origin of the sequences in question (genome projects, transcriptome analyses, or individual experiments) as well as the gene sequences and their localization and deleted incorrect and multiple sequences. In this process, we also took advantage of further alignment analyses and motif analyses explained in Section 2.5. To avoid accidentally deleting correct protein sequences, we performed the exclusion step by step and checked it repeatedly with further phylogenetic trees. For the final tree, we used 2249 sequences from 639 organisms belonging to different taxa. To balance between superfamilies with varying quantities of annotated genes, we included at most only five organisms from the same order and two of the same family for the phylogenetic tree. All sequences considered for this phylogenetic tree are listed in Appendix A. Phylogenetic trees were further processed with FigTree v1.4.4. 

### 2.4. Identification of Cryptochrome/Photolyase Subfamilies

Using the unrooted phylogenetic tree, we identified the different CRY/PL family members. For this purpose, we used already known CRY/photolyase sequences as reference (Table 1). 

### 2.5. Characterization of CRY/PL Family Sequences via Their Protein Motifs

To further confirm the accuracy of each subfamily, we carried out an additional motif analysis using the Annotate & Predict tool of the Geneious Prime software 2021.2.2. Thereby, we searched for all previously known CRY/PL motifs in all sequences found. These motifs were extracted or derived from the results of previous publications in which different CRY/PL subfamilies were characterized. A detailed overview of all motifs, their references and our search parameters for individual motifs are given in Appendix A. In addition, using subfamily alignments and generating consensus sequences, we identified novel protein motifs for previously not well-characterized CRY/PL subfamilies including PCRY-like, PPL, and CRY4. We further identified novel motifs for clear distinction between different types of MCRY. To determine the specificity of the new motifs discovered in our study, we also searched all 2249 sequences examined for these new motifs.

### 2.6. Assignment of CRY/PL Sequences to Their Subfamilies and Organisms

Based on tree branches and present motifs, we classified the protein sequences into their respective CRY/PL subfamilies We then investigated, which CRY/PL subfamilies the studied organisms possessed. This distribution was then incorporated into the phylogenetic taxon of each organism listed in Appendix A. The complete list of all organisms tested, and their CRY/PL repertoire can be found in Appendix A. The overall distribution and the frequency of presence of each CRY/PL family member within a taxon was then calculated. For organisms with questionable CRY/PL distribution, we also reviewed the available annotation and assembly qualities by BUSCO scores [41] and assembly contig-N50 values [42] to provide more clarity on the completeness of the results. 

### 2.7. Figures

Figures 1, 4, 5, 6, and 7 were created entirety or in part using BioRender.com, accessed between 1 August 2022 and 2 September 2022.

## 3. Results

### 3.1. Phylogenetic Tree

To assign the previously sequenced and annotated CRY/PL protein sequences to the appropriate subfamilies, an unrooted phylogenetic tree was constructed. The tree consisting of all identified and selected CRY/PL family members resulted in four main clusters: the 6-4-photolyase cluster (I), the CPDII cluster (II), the CPDI/III cluster (III) and the DASH cluster (IV) (Figure 2). The first 6-4 photolyase cluster consists of five CRY/PL subfamilies, including 6-4 PLs (pink, reference: 6-4 PL of Drosophila melanogaster) and four other animal CRY subtypes: MCRYs (red, reference: CRY1 of Mus musculus), DCRYs (blue, reference: CRY of Drosophila melanogaster), CRY4s (purple, reference: CRY4 of Danio rerio), and a small group consisting of sequences from the cnidarian class Anthozoa ACRYs (dark blue). The 6-4 PLs and CRY4 are from the same branch and thus closely related. The next close relative of 6-4 PLs are MCRYs, whereas DCRYs are more distant. The ACRY sequences form their own branch along the DCRY branch. 

The CPDII cluster (light green, reference: *D. melanogaster* PHR) contains CPDII photolyases. The DASH cluster consists of DASH-CRY sequences (orange, reference: DASH-CRY of *S. sp. PCC 6803*) and the plant photolyase PPL sequences (yellow, reference: *A. thaliana* PHR2). We find four CRY subfamilies in the CPDI/III cluster. It contains both CPDI (dark brown, reference: *E. coli* PHOTOLYASE) and CPDIII (brown, reference: *A.fabrum* PHOTOLYASE) and also includes the subfamilies of PCRY-like (turquoise, reference: *P. tricornutum* CryP), and PCRY (dark green, reference: *A. thaliana* CRY2). In addition to the obvious branches, we also found sequences that could not be clearly assigned to the above groups (grey). We find CRY/PL members from the sea louse *Caligus rogercresseyi*, the cat flea *Ctenocephalides felis* and the flatworm *Macrostomum lignano* between the CRY4/6-4 PL branch and the DCRY branch. A CRY/PL sequence of the tunicate *Oikopleura dioica* clusters between DCRY and ACRY. CRY/PL sequences belonging to the four studied Porifera and sequences from the diatom *Pseudo-nitzschia multistriata* and from the red algae *Galdieria sulphuraria* locate between the ACRY and DASH-CRY branches. Next to PCRY we find another black branch with CRY/PL members from five different red algae we cannot assign to known CRY/PL subfamilies. Between the CPDII and the CPDI/III clusters, we also located CRY/PL sequences belonging to different euglenozoans. The detailed tree in higher resolution can be found in Appendix A, and a rooted view for better classification can be found in Appendix A.

### 3.2. Motif Analyses

To verify the accuracy of the phylogenetic tree, we performed a motif analysis via the Annotate & Predict tool in the Geneious Prime software using all so far known CRY/PL motifs. In addition to the previously known motifs, we also found thirteen new motifs through alignment and consensus sequence analyses. A detailed description of all examined CRY/PL motifs (known and new) can be found in Appendix A. All alignments including the representation of their protein motifs are given in Appendix A.

#### 3.2.1. Novel Motifs Enable Strict Differentiation between 6-4 PLs and CRY4s

6-4 PLs are characterized by their α helices (α8, α10, α12, α14, α15, α16, α17, α18) and the sulfur loop (sl). In addition, 76% of them also possess the plant N-terminal motif (pn). All motifs found in 6-4 PLs are also abundant in the other CRY/PLs belonging to the 6-4 cluster, especially in MCRY and in CRY4 sequences. The motif analysis revealed that the most common motif among all members of the CRY/PL family is the α helix10 (α10) (Figure 3). This is strongly conserved across all CRY/PL members. Only in CPDI photolyases we found it rarely (38%). In our analysis, we found a novel 6-4 PL specific motif (YIYEPWKAP) in their C-terminal tail. It is present in 81% of 6-4 PL sequences and 73% of DCRY sequences, while only 44% of MCRYs and 21% of CRY4 sequences feature it (not shown in Figure 3 as < 50%). CRY4s are very similar to the 6-4 PLs in terms of their motifs. In addition to the previously described 4c and 4Y motifs, we also identified two new CRY4 characteristic motifs: 4a and 4b. All, except 4a, are almost absent in other subfamilies. Only 4a occurs in some MCRY sequences with a probability of 36% (not plotted).

#### 3.2.2. MCRYs Are Highly Conserved

MCRY is the most conserved subfamily in our analysis as all major motifs, especially the 6-4 PL specific helices, occur very frequently. All α helices and the sulfur loop occur in MCRY subfamily with higher frequencies than in the 6-4 PL subfamily. In addition, we also found the MCRY specific motifs second pocket up (m2pu), second pocket low (m2pl), phosphate motif (mpm), protrusion loop (ml), C-terminal lid (mcl) and C-terminal tail (mct). These MCRY-specific motifs, on the other hand, are not as strongly conserved, but feature characteristic differences between duplications and species. We will discuss these in more detail later. 

#### 3.2.3. DCRY and ACRY Differ Greatly in Their Motifs

The α10 motif is the most conserved motif from the Drosophila-type DCRY subfamily (90% frequency). In addition, most DCRY sequences feature further 6-4 PL motifs such as α-helices α14 (78%) and α17 (69%) and the plant N-terminal motif (pn) (76%). Besides the known DCRY typical sequences second pocket up (d2pu), second pocket low (d2dpl) and the C-terminal lid (dcl), we discovered in our analysis a new DCRY specific motif (dm1, HTLWxP), which appears in 70% of the investigated DCRY sequences. Most ACRY sequences are duplicated and, with about 1000 amino acids, form the largest members of the CRY/PL family in terms of protein size. Like MCRY and DCRY, ACRY also feature 6-4 PL motifs (α8, α10, α14, and α16) as well as the plant N-terminal motif (pn). The above motifs occur at least once (very often twice) in the protein sequence. The second plant N-terminal motif and the second α14-motif occur in all ACRY sequences (Figure 3). 

#### 3.2.4. CPDI and CPDIII PLs Are Relatively Similar, Whereas CPDII PL Differ Greatly

Many CPDII sequences also contain the α10 motif (90%), though the subfamily-typical motifs CPDII conserved1 (cc1) and CPDII mitochond. target (mts) occur more frequently. The mts motif is the most common, accounting for 98%. CPDII conserved2 (cc2) and CPDII conserved3 (cc3) are not as highly conserved with a frequency of 85% and 64%. CPDI photolyases are very different from the other members of the CRY/PL family. The two highly conserved motifs α10 and the plant N-terminus occur in only 38% and 10% of all CPDI sequences, respectively. Similarly, the CPDI/III-specific motifs are hardly found in other members of the CRY/PL subfamily, except CPDIII PLs and DASH-CRYs. While CPDIII conserved (c3) and CPDI/III conserved2 (c3-2) are relatively highly conserved in the subfamily (75% and 78%), CPDI/III conserved1 (c1/3-1) is less conserved with a frequency of 60%. In contrast most of the CPDIII sequences possess plant N-terminal (78%) and the α10 (67%). The c3 motif is most conserved as all tested CPDIII sequences possess it, while the c1/3-1 motif is only found in 22% CPDIII sequences (not shown).

#### 3.2.5. PCRY-Likes Lack PCRY-Specific Motifs, but Possess Three Novel Characteristic Motifs

The most common motif in the PCRY sequences is DAS motif1 (pd1) with a frequency of 97%. The plant N-terminal motif occurs at 93% in all analyzed sequences. The other PCRY-typical motifs, plant CRY conserved (pc) and plant CRY1 conserved (pc1), occur with 90% frequency. Only plant DAS motif2 (pd2) occurs less frequently at only 55%. None of the PCRY-specific motifs occur in PCRY-like sequences. PCRY-like cryptochromes possess only three known CRY/PL motifs. These are α10 (90%), α14 (85%) and the plant N-terminal (77%) (Appendix A). Both are generally among the most frequent motifs in the entire CRY/PL family. In our study, we identified three novel PCRY-like specific motifs (pl1, pl2, and pl3). PCRY-like 1 (pl1) consists of only four amino acids VGLR and occurs with a frequency of 74% in the PCRY-like subfamily. PCRY-like 2 (pl2, sequence: WRDLAY) is the second most common with 83%. The third motif PCRY-like 3 (pl3, sequence: MWQNxG) is the most conserved with 97%. All three motifs barely occur in other subfamilies.

#### 3.2.6. Novel PPL Motifs Allow Strict Discrimination between DASH-CRY and PPL

The DASH-CRY typical motifs are D-R salt bridge2 (db2, 93%), CPD-lesion repair (dlr, 73%) and D-R salt bridge1 (db1, 75%). In addition, DASH sequences have both the 6-4 photolyase-specific motifs α10 (73%) and α8 (52%), and c3, which is typical of CPDI/III photolyases, at a frequency of 88%. The DASH CPD lesion repair (dlr) also occurs in the plant photolyases PPL subfamily with a probability of 80%. All PPL sequences possess the α10 motif. Furthermore, we found by alignment and consensus sequence analyses three novel PPL specific motifs ppl1 (85%, ESxSxxPVYCFDPR), ppl2 (75%, BSJYGANFSCKI), ppl3 (85%, FFRFxTxK) (Figure 3, Appendix A). Except for ppl1 that also occurs with 2% frequency in the DASH subfamily, the motifs ppl2 and ppl3 are highly subfamily-specific allowing strict differentiation from DASH-CRY sequences clustering very closely in the phylogenetic tree.

### 3.3. Distribution of CRY/PL Family Members within the (Super-)Kingdoms

Based on the phylogenetic tree and motifs, individual members of the CRY/PL family were classified into their subfamilies. We assigned the classified sequences to the corresponding organisms, and then examined the organisms having specific CRY/PL subfamilies (Table 2). The detailed table showing each species and its CRY/PL family members can be found in Appendix A. In the tested archaea, we detected all four photolyases (6-4, CPDI, CPDII and CPDIII photolyases). However, they each possess only one photolyase type. The *Euryarchaeota archaeon* is the only archaeon studied having two CRY/PL family members in parallel, a 6-4 and a CPDIII PL. In the investigated bacteria, we detected all four photolyases, with CPDI being the major photolyase in this domain. While some of them have only one type of photolyase, others have two or even all three types of photolyases. In addition to the aforementioned CRY/PLs, about 30% of the bacteria studied, particularly most cyanobacteria, also possess the DASH-CRY. 

The eukaryotes studied were divided into four groups according to their kingdoms: unicellular eukaryotes, Viridiplantae, Fungi, and Metazoa. All previously described members of the CRY/PL family are also found in eukaryotes. In unicellular eukaryotes, an additional member of the CRY/PL family, PCRY-like, appears for the first time. In addition, we found the plant photolyase PPL in some unicellular eukaryotes belonging to the class Rhodophyta (red algae, Appendix A). Many unicellular eukaryotes, including red algae and kinetoplastids, also feature other CRY/PL members that cannot be assigned to the known subfamilies (Appendix A). Photolyases 6-4 and CPDII, along with DASH-CRY, PCRY, and PPL, are the major CRY/PLs in the Viridiplantae (plants). Most plants possess these five types, and through gene duplications they sometimes have multiple members of the same CRY/PL subfamily. Several green algae also possess a PCRY-like (approximately 11% of the Viridiplantae examined). *Chloropicon primus* possesses both PCRY and PCRY-like. We detected a CPDIII photolyase in the green algae *Ostreococcus tauri* and a CPDI photolyase in *Tetrabaena socialis* (Appendix A). No CPDI, CPDIII, or PCRY-like are found in the higher plants studied.

The main CRY/PL of Fungi is the CPDI photolyase. However, the dimorphic fungus *Mucor lusitanicus* appears to be the exception among the four microsporidia that were examined as CPDI could not be detected. Moreover, Microsporidia are the only fungi possessing CPDII photolyase. In addition to CPDI, many of the fungi examined also possess a 6-4 PL and/or a DASH-CRY. With a total of 497 individuals, the metazoan kingdom is the largest group in this study. We identified all previously described CRY/PL family members except the plant specific PCRYs and PPLs in metazoans. MCRY and DCRY occur only in the metazoan kingdom. 86% of the metazoans possess MCRY, the major CRY/PL of metazoans. CPDII (76%) is the major photolyase, followed by 6-4 PL (47%). DASH-CRY is present in about half of the metazoans studied. DCRY and PCRY-like are found only in certain metazoans (each 18%). Some metazoans also have their own CRY/PL types, such as ACRY (2%) and CRY4 (25%). CPDI and CPDIII photolyases are found in only few animals. 

### 3.4. Prevalence of CRY/PL Family Members within the Metazoa

To get an overview of the distribution of CRY/PL family members in each strain within the metazoans, we subdivided the animals from Appendix A within their strain and examined their CRY/PL family member distribution in more detail (Figure 4). We could identify all previously described CRY/PL family members except the plant specific PCRY and PPL in metazoans. Due to the limited number of sequenced and annotated species of some phyla, drawing conclusions is difficult. Nevertheless, we observe strong variability in CRY/PL composition between individual phyla. Some phyla appear to possess only one of the CRY/PL subfamily members. For example, in *Trichoplax adhaerens* (Placozoa) we only detected a CPDIII photolyase, while in all Ctenophora (comb jellyfish) studied we detected only the 6-4 PL. Nearly all animals examined from the phylum Porifera (sponges) possess only one species-specific CRYs. The demosponge *Amphimedon queenslandica* is the only exception, possessing an additional CPDII photolyase. All platyhelminths examined possess the CPDII photolyase. About 22% of them also possess the 6-4 PL. Nematodes examined possess either CPDII- or CPDI-photolyase only, with CPDII-photolyase being the most abundant. In the water bear *Ramazzottius varieornatus* (Tardigrada) we found only a CPDII photolyase. In the arrowworm *Spadella cephaloptera* (Chaetognata) we also identified only the DCRY. All other phyla possess a higher number of CRY/PLs. 

MCRYs appear for the first time in cnidarians. All studied cnidarians belonging to the subclass Hexacorallia possess the MCRY (Appendix A).2 Additionally, they also possess the 6-4 PL, CPDII PL, DASH-CRY, and an Anthozoa-specific cryptochrome ACRY. Most animals belonging to the phylum Echinodermata possess nearly all CRY/PL family members, including DCRY. They only lack the photolyases CDPI and CPDIII, and the cryptochrome PCRY-like. The chordates also possess almost all CRY/PLs found in the echinoderms. However, DCRY is completely absent in chordates (Figure 4). MCRY is their most abundant CRY/PL, but DASH-CRY and the photolyases 6-4 and CPDII are also very common. In addition, many chordates (41%) possess the CRY4 and a smaller number possess PCRY-like (26%, Appendix A). 

Like Echinodermata, which belong to Deuterostomia, animals belonging to Protostomia also possess DCRY. Among the Protostomia, we first find DCRY in *Lingula anatina* (Brachiopoda) and in the arrowworm *S. cephaloptera* (Chaetognata). We also detected DCRY sequences in rotifers, annelids, and mollusks. While only rotifers belonging to the subclass Monogonata have DCRY, all annelids tested and nearly all mollusks (90%) possess it. The rotifers possessing DCRY also have a 6-4 PL, while all other rotifers mainly have a CPDII photolyase and most also a DASH-CRY (Appendix A). Annelids and mollusks share a similar CRY/PL distribution, by expressing MCRY, DCRY and CPDII photolyases as their main CRY/PLs, followed by 6-4 PL and DASH-CRY. 33% of annelids and 20% of mollusks also possess PCRY-like. MCRY, DCRY, 6-4 PL, and CPDII PL are very abundant in arthropods, DASH-CRY, CPDI, and CPDIII photolyases are found in only a few of them. In the following, we will take a closer look into the distribution of CRY/PL family members within the two phyla Chordata and Arthropoda.

### 3.5. Prevalence of CRY/PL Family Members within Chordata

Most chordates studied (303 of 306) have a skull of hard bone or cartilage and thus belong to the craniates (Figure 5A). Only, the Cephalochordata (lancelets) and Tunicata lack such a scull. The tunicates are very special and the one species studied here, the palegic tunicate *O. dioica*, possesses only a CPDI PL and an additional CRY/PL family member, which cannot be clearly assigned, but clusters in the phylogenetic tree close to the DCRY (Figure 2, Appendix A). In contrast, the two lancelets studied have the original “standard” MCRY (MCRY st), CRY 4 and CPDII (Figure 5A). In all craniates, MCRYs are the main CRY/PL family members. The original MCRY underwent a gene duplication, making it present twice as MCRY1 and MCRY2 (Figure 5B). We identified three new MCRY-specific motifs that allow a clear distinction between the standard MCRY and the two chordate-specific MCRYs (Appendix A). Almost all the craniates studied, with only few exceptions, possess both MCRYs. Exceptions for this rule are the sea lamprey *Petromyzon marinus* (class: Hyperoartia) and sharks of the order Orectolobiformes (class; Chondrichthyes), which have MCRY1 only (Figure 5A). Also, few other chordate species lack one of the two MCRYs. Several species possess only MCRYs (many Chondrichthyes, some Gymnophiona (class: Amphibia), half of the Prototheria, few Metatheria, the only studied species of the Coelacanthimorpha—the coelacanth *Latimeria chalumnae*, and almost all Eutheria (47 of 48).

Among the Eutheria, only the sperm whale, *Physeter catodon*, possesses an additional CPDII photolyase. Similarly, CPDIIs are the only photolyases besides MCRYs found in see lamprey, the abovementioned sharks, caecilians (order: Gymnophiona), half of the Prototheria and most Metatheria. On the other hand, some craniates, such as frogs (Anura) possess 7 CRY/PL family members although they are closely related to the Gymnophiona, which have only MCRYs and CPDII (Figure 5A). DASH-CRYs and 6-4 PLs are present in 8 of the 17 Craniata phyla, respectively (not necessarily the same), PCRY-likes are present in 2 phyla (bony fishes belonging to the Neopterygii and frogs), and CRY4s are present in 40% of the frogs, all investigated turtles (Testudines), most birds (Aves and Neognathae) and one lizard, the green anole *Anolis carolinensis* (Lepidosauria) (further details see Figure 5A).

### 3.6. Prevalence of Members of the CRY/PL Family within the Phylum Arthropoda

Arthropods are the most successful and the most biodiverse phylum among the metazoans. Chelicerates, crustaceans, and hexapods form the major subphyla of arthropods. The main CRY/PL members of the arthropods are MCRY, DCRY and the photolyases 6-4 and CPDII (Figure 6). In addition, some arthropods, mainly crustaceans, carry the DASH-CRY, too. CPDI and CPDIII photolyases also appear occasionally, mainly in chelicerates. While the horseshoe crab *Limulus polyphemus* has all four arthropod-typical CRY/PLs, the spider crab *Nymphon striatum* has only the CPDIII photolyase. The chelicerates of the class Arachnida differ greatly within their subclasses and their orders. While the Aranae have the typical four arthropod CRY/PLs, scorpions only have the MCRY. Acaris of the order Ixodida (ticks) also have MCRY only, while the Mesostigmata have the CPDII photolyase only. Acaris of the order Sarcoptiformes, on the other hand, possess only the CPDI photolyase. 

In Ostracoda we find MCRYs, DCRYs and CPDII photolyases. Other animals belonging to the crustacean subphylum Thecostraca, Branchiopoda and animals of the Cohort Eucarida (subphylum Malacostraca) possess in addition to MCRYs, DCRYs and CPDII PLs also 6-4 PLs and DASH-CRYs. In contrast, all animals from other Cohorts of Malacostraca lack DCRYs and DASH-CRY. The other CRY/PLs mentioned above are not always present either, except for MCRYs, which are found in all Malacostraca. In the amphipod *Hyalella azteca*, we found a CPDI photolyase in addition to MCRY and CPDII. 

The four arthropod-typical CRY/PLs are found in all subgroups of the Hexapoda subphylum. Even though all four CRY/PLs are found in the individual hexapod groups, they are not equally present in all animals. For example, the animals of the order Hymenoptera have at most only the CPDII photolyase in addition to the always present MCRYs, whereas all the lepidopterans studied have all four arthropod-specific CRY/PLs simultaneously. MCRY is the main CRY/PL of the hexapods. Only some insects of the cohort Holometabola (order Diptera) do not have MCRYs, including the fruit fly *D. melanogaster*. In Collembola and in individual insects of the Cohort Paraneoptera and Holometabola, also DASH-CRYs are present.

## 4. Discussion

### 4.1. Genesis of the CRY/PL Family Members

In this study, we investigated the distribution of CRY/PL, as well as its emergence and loss during evolution. Although we mainly focused on metazoans, we additionally included some selected archaea, bacteria, protists, plants, and fungi as references. Regarding metazoans, we included all species with sequenced and annotated genes so far, though limited to a maximum of five per order and two per family. Still a possibility exists that that we overlooked few CRY/PL protein sequences not yet annotated. For this reason, we emphasize that this study only reflects the current state of knowledge, especially of annotated protein sequences until February 2022. Nonetheless, the high agreement of our results with the available literature reinforces our findings [19,20,43,44]. Furthermore, the additional protein motif analyses performed corroborate the accuracy of the individual CRY/PL members and their assignment to the subfamilies.

#### 4.1.1. Photolyases Are Ancient Molecules Found in All Three Kingdoms

Photolyases are ancient DNA repair enzymes thought to have evolved before atmospheric oxygenation and the formation of the ozone layer [19,20,45,46]. They were crucial at those times as intense UV radiation could reach the surface of the Earth without attenuation by the ozone layer [45,47]. Indeed, we found all four photolyases in Archaea and Bacteria with CPDII being the most abundant in Archaea and CPDI the most abundant in Bacteria. Most procaryote species possess only one photolyase, but *E. archaeon* has the 6-4 and CPDIII photolyases together, and several bacteria possess the 6-4 and the CPDII photolyases together with the CPDI photolyase. Interestingly, the CPDI and CPDIII photolyases never occur simultaneously, and this also applies to all eukaryotes examined in this study. Zhang et al. (2013), assumed that a prokaryotic 6-4 photolyase with an iron-sulfur cluster (FeS-BCP) is the first common ancestor of photolyases [48]. In contrast, the recent study by Xu et al. (2021), considers this unlikely, as functional changes from 6-4 to CPD repair mechanisms must then have occurred multiple times during evolution [49]. Furthermore, they emphasize that CPDs are the major UV-induced DNA lesions and therefore it is more likely that CPD photolyases are the most ancient. They consider a new protein class of short photolyase-like (SPL) proteins similar to CPD photolyases, as possible origin of photolyases and thus of the CRY/PL family [49]. Our study confirms the special role of CPD photolyases as they are the most abundant across taxa. We also confirm the common hypothesis that photolyases are the origin of the CRY/PL family as we find all photolyases in all three domains (Table 1, Figure 7).

#### 4.1.2. DASH-CRY: The First Cryptochrome or Still a Photolyase?

DASH-CRYs are probably the first cryptochromes within the CRY/PL family [25]. Among archaea, only the extremely halophilic archaeon *Halorhabdus utahensis* (not included in our study) appears to have a DASH-CRY [19], but DASH-CRYs are widely distributed in the domains of Bacteria (especially Cyanobacteria) and Eukaryota. In addition to the DASH-typical motifs, DASH-CRYs possess the c3 motif, which is found only in CPDI/III photolyases and their relatives. Because of this structural similarity and because of their proximity to the CPDI&III branch in the phylogenetic tree (Figure 2), we hypothesize that DASH-CRYs are closely related to CPDI/III photolyases and have evolved from a common precursor, forming sister clades. The fact that at least some DASH-CRYs are also able to repair CPD lesions supports our hypothesis [25]. Furthermore, DASH-CRYs form a bridge between CPDI/CPDIII photolyases and other members of the CRY/PL family, especially the 6-4 PLs since most DASH-CRYs additionally possess the major motif of the CRY/PL family, α10, and the α8 motif, which we find mainly in 6-4 PLs and in MCRYs. Whether the presence of different CRY/PL subfamily motifs has arisen through convergent evolution or rather through homologous recombination between a CPDI or CPDIII photolyase and a 6-4 PL in an early bacterium is still unclear. DASH-CRYs are versatile in their function and, particularly in Fungi, they appear to be involved in gene regulation and development [27,50,51]. In the green algae *C. reinhardtii*, DASH-CRYs seem to play a role in the photosynthetic machinery [52], while biochemical studies show that many DASH-CRYs are capable of CPD repair in ssDNA [25,53,54,55]. Apparently, DASH-CRYs partially retained the DNA repair activity of their photolyase precursor, but simultaneously acquired new photoreceptive functions, particularly in light-dependent organisms.

#### 4.1.3. Plant CRY-Like (PCRY-Like)

PCRY-likes (plant CRY-likes) belong to the next CRY/PL member that appeared in early evolutionary history, initially in simple eukaryotes (Figure 7, Appendix A). A PCRY-like was first described in the diatom *P. tricornutum* and in other marine organisms [28,29,30]. Although named PCRY-likes, they bear little resemblance to PCRYs in terms of motifs (Figure 3). The plant N-terminal (pn) and the α-helices α10 and α14 are the only common motifs, but since these are anyway most common in all CRY/PLs they are not helpful in PCRY-like assignment. However, the pl1, pl2 and pl3 motifs identified in our study allow a precise assignment. In particular, the pl3 motif (MWQNxG) occurs with 97% in all PCRY-like sequences (100 out of 103 sequences), but in none of the other 2144 sequences belonging to other CRY/PL subfamilies. Interestingly, nearly all organisms that possess a PCRY-like additionally possess a DASH-CRY (Appendix A). Only five organisms do not possess both PCRY-like and DASH-CRY simultaneously. Apparently, PCRY-likes and DASH-CRYs are linked in an interplay, which has not been investigated so far. Moreover, PCRY-likes seem to have a unique position in the CRY/PL family, as their occurrence has little to do with phylogenetic relationship between species, but rather with their natural habitat. While we find PCRY-likes in isolated species of fish and frogs as well as in water-associated crustaceans, we do not find it in their close relatives living in terrestrial habitats. Also, in plants and in other unicellular eukaryotes they occur exclusively in aquatic species, regardless of their phylogenetic relationship. Horizontal gene transfer (HGT) seems to play an important role in PCRY-like distribution. HGT is already a known factor in the CRY/PL family evolution occurring more frequently in aquatic organisms [20,56,57]. 

#### 4.1.4. Plant CRY (PCRY) and Plant Photolyase (PPL)

Continuing the evolutionary history, we find that kingdom-specific CRY/PL members have evolved in multicellular organisms. Thus, plants additionally possess the plant CRYs (PCRYs) as well as the plant photolyases (PPLs). Both are mainly found in higher plants and some red algae (Figure 7). Already Charles Darwin described the ability of plants to perceive blue light [58], and thus a “cryptic” blue-light photoreceptor (namesake for cryptochrome) was suspected in them. Indeed, the first cryptochrome ever, encoded by the gene *HY4*, was discovered in the thale cress *A. thaliana* [59]. After characterizing and finding its similarity to photolyases, its involvement in the plant circadian clock was revealed [11,60,61,62]. PCRYs are thought to be closely related and derived from CPDIII photolyases [19,30,44,63]. Although the PCRYs are also very close to the CPDIII photolyase sequences in our phylogenetic tree (Figure 2), we do not find a direct link between CPDIII and PCRY sequences in terms of protein motifs. Moreover, a duplication of the PCRY gene led to the emergence of two gene products PCRY1 and PCRY2 with specialized functions in higher plants. A detailed analysis of these PCRY subgroups can be found in the reviews by Liu et al. (2016) and Wang et al. (2018) [64,65].

Plants as sessile and photoautotrophic organisms are at high risk of UV damage. To protect themselves they express besides the known photolyases (especially 6-4 and CPDII photolyase) also the plant specific photolyases PPL. Although PPLs have no sequence homologies to CPD photolyases, studies in *C. reinhardtii* suggest a CPD photolyase repair activity for chloroplast and nuclear DNA [34]. In contrast to PCRYs, which are related to CPDIII photolyases, PPLs are related to and likely evolved from DASH-CRYs [19]. Our study further supports this hypothesis, as PPLs particularly cluster very close to plant DASH-CRYs (Figure 2). Most likely, PPLs have evolved from a gene duplication of a DASH-CRY gene within a plant ancestor. A characteristic of the PPLs is that their FAD-binding domains are truncated and less conserved than in the other subfamilies [19]. Other characteristic motifs were previously not reported for PPLs. We identified for the first time three PPL-specific motifs, PPL1, PPL2, and PPL3, which allow differentiation from other members of the CRY/PL family, in particular from DASH-CRYs (Figure 3, Appendix A). 

### 4.2. The Rise of Animal Cryptochromes

All animal CRYs are closely related to and derived from the 6-4 PLs (Figure 2 and Figure 3). Like PCRYs, most animal CRYs, except CRY4s, seem to be involved in the circadian clock. Specifically, MCRYs and DCRYs are known key molecules of the circadian clock, though in different ways. So far, different circadian clocks are known, in which either only one of them or both are involved [43,66]. The large number of metazoans examined in our study allows us to analyze the evolution of the various animal CRYs in detail. Before doing so, we need to consider the evolutionary relationships between metazoans in more detail, as these remain controversial, particularly for the early metazoans Ctenophora, Placozoa, and Porifera [67,68,69,70,71]. Although Porifera were long thought to be the most primitive phylum, recent studies suggest that Ctenophora are probably more basal [72,73]. In addition, their relationship to Placozoa remains unclear [74,75]. Remarkably, each of these phyla exhibits distinct CRY/PL family members.

#### 4.2.1. Original Metazoans Strongly Differ in Their CRY/PL

All tested Porifera species possess a species-specific cryptochrome, only the demosponge *A. queenslandica* possess a CPDII photolyase in addition. The species-specific CRY was previously described as eye type or type 0 CRY [1,44,76]. In these studies, as well as in our study, the sponge CRYs cluster near to the animal CRYs, especially near DCRYs (Figure 2). But in our study, they do not form a distinct group, even when spatially grouped together. This could be due to the early separation of single species of this evolutionarily primitive phylum. Moreover, we could not find any type-specific motifs in the sponge CRYs. Some of them are only characterized by the presence of sporadic α-helices of the 6-4 photolyase (α8, α10, sl & α18, see also Appendix A). Studies on the sponge *Suberites domuncula* indicate that these CRYs are photosensitive and involved in various processes such as phototactic behavior, phototransduction, and circadian rhythmicity [76,77]. This suggests that the common ancestor of the Porifera and Bilateria already possessed an animal CRY. *T. adhaerens* is one of the three recognized species of Placozoa and the only one with a sequenced and annotated genome. *T. adhaerens* possesses the smallest animal genome and is the simplest free-living multicellular animal [78]. This could be the reason why it has only one CPDIII photolyase, which can be considered as all-rounder featuring PCRY-, MCRY-, 6-4- as well as CPDI/III-specific motifs (Appendix A). In contrast, all four animals belonging to the phylum Ctenophora are specialized to only 6-4-PLs. Although we find no evidence for animal CRYs in both phyla, their exclusion should be considered with reservation due to the few animals sequenced.

#### 4.2.2. Cnidarian: Animal CRY Expansion > Birth of MCRY and ACRY 

Cnidarians being relatively close to bilaterians are not considered to be among the most primitive animals [69,70,79]. We found more than 12 sequenced cnidarians allowing us to study individual classes and subclasses in more detail. In cnidarians, which are known for their remarkable plasticity of morphology and life we find a kind of “CRY/PL expansion” (Appendix A). While we found only the DASH-CRY in the freshwater polyp *Hydra vulgaris* (class: Hydrozoa) and the CPDII photolyase in other cnidarians of the class Myxozoa, we detected several CRY/PL family members in the class Anthozoa, specifically in the subclass Hexacorallia. Animals of this subclass possess many CRY/PL family members, including the already known CPDII and 6-4 photolyases as well as the DASH-CRYs. In them, we also discovered for the first time the major animal cryptochrome MCRY (Figure 4b and Figure 7, Appendix A). Previous studies have already reported the presence of MCRY in cnidarians [40,80]. Their MCRY sequences differ from the other sequences and barely possess any MCRY-typical motifs, except the MCRY2 specific motif II (Figure 3, Appendix A).

Hexacorallia also possess ACRY, which is not present in any of the other organisms studied (Figure 4b and Figure 7). ACRY sequences are characterized by duplications and the basic sequence is appended up to five times. For example, the sequences of *Nematostella* XP_032237794.1 and XP_032237792.1, which consist of 1985 and 2482 amino acids, have 4-5-fold repeats. In our analysis, we only considered CRY/PL homologues with single or double repeats. While in former studies ACRY was classified to be a type I cryptochrome [80], which is a synonym for DCRY, in our study it forms a separate clade close to DCRY (Figure 2). Moreover, except the ACRY of the mountain star coral *Orbicella faveolata*, having the DCRY-specific motif d2pl (Figure 3, Appendix A), all other ACRY lack any DCRY specific motifs. ACRY and DCRY though forming separate subfamilies, possibly share a common ancestor (Figure 2, Appendix A). Whether ACRY, like the other animal CRYs, is also involved in the circadian clock is uncertain. Gornik et al. (2021), ruled this out in their studies as they observed a light-induced rather than rhythmic expression in ACRY [40]. This conclusion should be taken with caution because for example one major clock gene of *D. melanogaster*, *cycle*, is also not rhythmically transcribed but is kept in sync by its partner CLOCK, which is rhythmically transcribed [81]. Thus, rhythmic transcription is no prerequisite for a clock gene. 

#### 4.2.3. MCRY Is More Ancient (Already in Planulozoa), DCRY Is Only Found in Bilateria

Besides cnidarians, also animals belonging to the subkingdom Bilateria possess the MCRY. This suggests that the origin of the MCRY occurred before the separation of the Cnidaria from the Bilateria, i.e., in the common ancestors of the Planulozoa [82]. DCRY, on the other hand, seems to have evolved later. So far, we have detected DCRY only in animals of the subkingdom Bilateria (Figure 4b and Figure 7). Here it occurs in both protostomes and deuterostomes, suggesting its emergence before the split of the two clades. In general, MCRY subfamily members are highly conserved across phyla. In our study, the abbreviation MCRY derives from mammalian-type CRY, but we may even call it the metazoan-type CRY. Indeed, MCRY is the most abundant cryptochrome in the animal kingdom, with a frequency of 86%. While CPDII photolyase is the second most abundant with 76%, the other animal cryptochrome DCRY is found in only 18% of animals (Table 1). 

#### 4.2.4. CRY4: The Chordate-Specific Magnetoreceptor

Some chordates, among them bony fishes, amphibians, turtles, lizards, and birds, possess an additional cryptochrome, CRY4. The phylogenetic tree clearly reveals that CRY4 is closely related to, and probably derived from, the 6-4 PL clade (Figure 2). It shares strong similarities with 6-4 PLs, though minimal amino acid changes, particularly the conserved Y319 in CRY4 proteins (4Y motif, Figure 3), seem to alter its photochemical properties and its mode of action [83]. In addition to the two known CRY4-specific motifs 4c and 4Y, we also identified two novel protein motifs enabling strict differentiation between 6-4 PLs and CRY4 (4a & 4b, Figure 3, Appendix A). Based on further analysis, we found that the 4a motif is also very abundant in MCRYs, especially in MCRY1 sequences. Some animals examined possess both CRY4 and 6-4 PLs (bony fishes, amphibians, lizards, and turtles), while other possess only the 6-4 PL or only the CRY4 (Appendix A). Whereas only single fishes of the Infraclass Neopterygii have CRY4, almost all have the 6-4 PL. During chordate evolution, the ratio reverses, as most birds have CRY4 and less 6-4 photolyase. The lancelets Leptocardii and birds not belonging to the infraclass Neognathae retain CRY4 only (Figure 5A). Besides birds, insects such as the monarch butterfly *Danaus plexippus*, migrating many hundreds of kilometers annually, are also renowned for magnetoreception [84]. CRY4 is however restricted to chordates. Since CRY4 is very similar to the 6-4 PL, but also to DCRY and MCRY, these might also function as magnetoreceptors by minimal modifications. Studies in cockroaches [85,86], fruit flies [87,88,89] and butterflies [90,91] already indicate their involvement in insect magnetoreception. 

#### 4.2.5. Animal CRYs in Protostomia: Spiralia & Ecdyzosoa

In contrast to the deuterostomes, DCRY is widespread among the protostomes. For the first time we see DCRY in the brachyopod *L. anatina*. Almost all studied animals belonging to the superphylum Spiralia including animals of the phyla Chaetognatha, Mollusca, Rotifera and Annelida contain DCRY. Platyhelminthes seem to be an exception in Spiralia as most of them only possess a CPDII PL (some also a 6-4 PL in addition). In the flatworm *Macrostomum lignano* we found another CRY/PL family member that is unassigned. Since it shares some common motifs with the 6-4 PL and the animal CRYs and is positioned between them in the phylogenetic tree, we assume that it might be a modified form of the animal CRY. The superphylum Ecdyzosoa consists of the phyla Arthropoda, Nematoda and Tardigrada, among others. While we identified both animal CRYs MCRY and DCRY in Arthropoda, neither the Tardigrada nor the Nematoda possess them. In the water bear *R. varieornatus* (Tardigrada) and in most nematodes, we found only the CPDII PL. This raises the question why some evolutionarily recent phyla lack the animal CRYs that appeared already in early Bilateria. Interestingly, some evolutionarily young phyla seem to have lost multiple CRY/PLs and retain only a few. We suggest that the era of development and retention of diverse CRY/PL family members was apparently followed by an era of reduction and concentration on single CRY/PLs. These reductions have not always occurred at the level of phyla, but often at the level of classes, families, or genera.

### 4.3. Selection and Restriction to Certain CRY/PL Family Members

It is questionable whether the first metazoans of the phyla Porifera, Ctenophora, and Placozoa, which also have only few CRY/PL members, can also be considered restrictive, because not enough animals of these phyla have been sequenced so far. This problem will certainly be solved in the next few years with new sequencing technologies. Another problem persists that many primitive animals filling the gaps in metazoan evolution are already extinct. Since the remaining animals of this original phyla have several million years of evolution behind them just like younger animals, they can only fill the gaps to a limited extent. Therefore, we only conclude that the studied animals of these primordial metazoan phyla have only a few CRY/PLs and, most importantly, that they do not possess MCRY, DCRY, DASH-CRY, or PCRY-like. However, we cannot declare whether they have always lacked these CRYs or whether they have lost them over time. In the following part of our discussion, we will look in detail at the CRY/PL restrictions in different metazoan groups.

#### 4.3.1. Echinoderms share multiple CRY/PLs, including DCRY

The large number of sequenced animals in both deuterostomes and protostomes allowed us to analyze the CRY/PL restrictions in more detail. Interestingly, many echinoderms (Deuterostomia) possess five metazoan specific CRY/PL family members being MCRY, DCRY, DASH-CRY and both photolyases 6-4- and CPDII (Figure 4). Exceptions are the three sea urchins *Mesocentrotus franciscanus*, *Evechinus chloroticus* and *Sterechinus neumayeri* in which we could only detect the CPDII PL. This probably results of incomplete annotation as these photolyase sequences are derived from a single study [92]. The absence of 6-4 PL in the feather star *Anneissia japonica* might be true as BLAST-search on the respective assembly with related 6-4 sequences did not yield reliable results. 

#### 4.3.2. The Original Chordates Have a Reduced CRY/PL Repertoire

Recently Kotwica-Rolinska et al. (2021), revealed that *Ptychodera flava*, a hemichordate, possesses DCRY [43]. Our study lacks hemichordates because most of them are not sequenced and annotated yet. An exception is the acorn worm *Sacoglossus kowaleyskii*, though having an annotated genome, we could not find any CRY/PL family members in it. The presence of DCRY in echinoderms and hemichordates strongly indicates that DCRY was lost only after the split of chordates. Besides the chordates listed in the phylogenetic tree, we also performed additional BLAST and motif analysis in all chordates sequenced so far and can confirm this loss with certainty. The pelagic tunicate *O. dioica* and the two lancelets *Branchiostoma floridae* and *Branchiostoma belcheri* are the only chordates in our study not belonging to the subphylum Craniata. In the lancelets, we find besides MCRY and CPDII photolyase also the chordate specific CRY4, a putative magnetoreceptor, which is likely derived from 6-4 PLs. Homologues of CRY4 were further identified and characterized in zebrafish [31], birds [93,94], amphibians [95], and in lizards [96]. Our study revealed that turtles possess CRY4 too. Interestingly, all these animals are supposed to have a magnetic sense [97,98,99,100]. Only for lancelets we could not find any studies on magnetic sense so far. However, the presence of CRY4 suggests a magnetic sense in them as well. Furthermore, the occurrence of CRY4 already in Craniata, suggests its early evolution in original chordates (Figure 5A).

The tunicate *O. dioica*, however, differs from all other Deuterostomia by having two CRY/PLs, one clustering with CPDI PL and the other near DCRY. Since hardly any metazoans and especially no chordates possess CPDI photolyase, we assume that this is either a contamination or a misclassification of the CRY/PL sequence. The former is supported by the fact that it is an aquatic species and contamination is common in these. However, in favor of the second assumption are its motifs, since this one, in addition to its α10, also has an MCRY protrusion loop (ml) motif, which would be more indicative of an MCRY. The second sequence (CAG5112545.1), has in addition to the plant N-terminus also the 6-4 PL specific motif, which occurs in 6-4 photolyases, but also in DCRY sequences. Given the proximity to DCRYs in the phylogenetic tree, this could indicate a DCRY relationship (Figure 3). Barring possible contamination and sequencing errors, this would suggest that this more primordial creature has greatly modified its CRY/PLs during evolution.

The sea lamprey *P. marinus* (class: Hyperoartia) possesses a CPDII photolyase in addition to MCRY. In almost all craniates, MCRY duplicates, MCRY1 and MCRY2, are present (Figure 5A). While lancelets still have an unduplicated standard MCRY, the MCRY in sea lamprey lies between the two duplicates (Appendix A). While we define it as MCRY1 based on its clustering (even if as an outgroup), it is still the only one to exhibit the MCRY2-specific motif (II) in the entire MCRY1cluster (Appendix A). Carpet sharks of the order Orectolobiformes (class: Chondrichthyes) have a similar CRY/PL distribution possessing an MCRY1 and a CPDII photolyase. But we find MCRY duplication in other cartilaginous fishes. Interestingly, all species except the great white shark *Carcharodon carcharias* that have MCRY duplication, have lost CPDII photolyase (Appendix A). Still, we assume that ancient chordates probably had many more CRY/PLs than today’s cartilaginous fishes and lancelets, as evolutionarily younger chordates such as bony fishes or frogs also possess many CRY/PLs. Possibly the lamprey and cartilaginous fishes of the early Devonian had more CRY/PLs and only lost them over time.

#### 4.3.3. CRY/PLs in Bony Fishes, Coelacanths and in the Lungfish

A CRY/PL increase or retention of multiple CRY/PLs is seen in bony fishes of the class Cladista. They have in addition to the two MCRYs also DASH-CRY, 6-4 and CPDII photolyases. In the class Actinopteri, the fishes belonging to the subclass Chondrostei again lack DASH-CRY, while most fishes of the subclass Neopterygii have DASH-CRY. Many of them also possess the PCRY-like, which appears to be closely linked to DASH-CRY. Fishes having the PCRY-like also usually have the DASH-CRY. The only exceptions are the white flower croaker *Nibea albiflora* and the allis shad *Alosa alosa*. The fact that the American shad *Alosa sapidissima* has both PCRY-like and DASH, but its relative *A. alosa* does not, is rather unexpected. Since *A. alosa* has a BUSCO score above 95%, we consider sequencing and annotation errors rather unlikely. Whether *A. alosa* lost PCRY-like and DASH later in evolution or rather the inclusion of them in *A. sapidissima* occurred subsequently should be further investigated. However, the distribution of PCRY-like in fish points out that although PCRY-like is present only in water-associated organisms, not all water-associated organisms necessarily possess it.

In fishes belonging to the subclass Neopterygii, the CRY4 emerges for the first time. While all three examined fishes of the Infraclass Holostei possess CRY4, only some fishes of the Infraclass Teleostei do. Fishes from the Cohort Euteleosteomorpha generally do not possess CRY4 (Appendix A). The only exception is the alpine whitefish *Coregonus* sp. “*balchen*”, where we found a CRY/PL sequence clustering with CRY4 sequences in the phylogenetic tree and possessing the CRY4-specific 4b motif (even if it is incomplete with only 182 amino acids). Assuming this is indeed a CRY4 it would mean that *Coregonus*, which occurs in alpine lakes such as Lake Constance, has a magnetoreceptor, but its close relative, the pink salmon *Oncorhynchus gorbuscha*, known for its migrations, does not. Since we also did not find MCRY2 in *Oncorhynchus* (Appendix A), we cannot exclude incomplete sequencing as a possible cause. Likewise, we cannot exclude the possibility that these animals have a peculiar CRY/PL mechanism. In the other classes, on the other hand, CRY4 is very often present. Many fishes possess up to seven CRY/PLs (CRY4, the two MCRYs, DASH-CRY, PCRY-like, 6-4 PL and CPDII PL). However, the catfishes of the order Siluriformes again have a reduced CRY/PL repertoire, with most species having only the CPDII photolyase in addition to the two MCRYs. In general, the combination of MCRY and CPDII photolyases seems to be quite common in the animal kingdom. Especially many animals with reduced CRY/PL repertoire have CPDII besides MCRY, if not only MCRY alone.

The coelacanth *L. chalumnae* was considered an extinct fossil for a very long time. The deep sea hidden fish was discovered to still exist only in 1939 [101]. Due to its habitat in the dark deep sea, the coelacanth seems to have lost all light-sensitive CRY/PLs. Specific changes in opsins were also detected in them as they only receive light around 480 nm in their environment [102]. Although CRY/PLs could also perceive this wavelength, there is probably no need due to the loss of other wavelengths, especially UV light. As a result, we could only find the light insensitive MCRY duplicates in *L chalumnae*. In contrast, its close relative the lungfish *Protopterus annectens* (subclass: Dipnomorpha) inhabiting rather small stagnant or slow flowing waters (in dry periods even without any water) is exposed to large amounts of light. *Protopterus* possesses again more CRY/PLs, besides the two MCRYs, it also has DASH-CRY and the two photolyases 6-4 and CPDII.

#### 4.3.4. CRY/PL Distribution among Tetrapods

The amphibians are subdivided into Anura (frogs) and the Gymnophiona (caecilians). Frogs that stay in water and on land have many (up to seven) CRY/PLs. In contrast, caecilians living hidden in the upper soil and litter layers seem to have lost many CRY/PLs due to their sheltered-from-light lifestyle and have left only both MCRYs and the CPDII photolyase. While the crocodiles possess the two MCRYs, CPDII and the 6-4 photolyases, the turtles have additionally also CRY4 and DASH-CRY. Although lizards and snakes belong to the same order of Squamata, they also differ greatly in their CRY/PL content. While lizards have similar amount of CRY/PLs like turtles, snakes are like caecilians having only MCRYs and CPDII PL. This again shows that for the CRY/PL repertoire not necessarily the degree of relationship, but other factors, especially environmental conditions of animals play a more important role. 

Besides the two MCRYs, most birds also have DASH-CRY and CPDII photolyase. Only one-third of the birds belonging to Infraclass Neognathae possess the 6-4 PL. Birds that do not belong to this Infraclass lack 6-4 PL at all. Instead, many birds, whether migratory or stationary, have CRY4. CRY4 was first discovered in zebrafish and was thought to be a circadian photoreceptor due to its high sequence similarity to DCRY and its rhythmic expression [31]. However, CRY4 has been shown to have a constant expression in zebra finch (*Taenopygia guttata*) retina, which is considered an important requirement for a magnetoreceptor [103]. Most probably, there are two different CRY4 isoforms as was recently discovered in the night-migratory robin *Erithacus rubecula*. One of them (CRY4a) is highly regulated during the migratory period and thus has a seasonal rhythm, but no diurnal rhythm [104], suggesting that it is a magnetoreceptor. In contrast, the other isoform CRY4b, showed a circadian regulated expression pattern [105], so that another (possibly clock-related) function cannot be excluded. 

#### 4.3.5. Highly Reduced CRY/PLs in Mammals

In the age of the dinosaurs, most mammals were small and nocturnal. Only after the extinction of dinosaurs about 66 Ma ago, some of them became diurnal [106]. A CRY/PL reduction, caused by nocturnal activity persisted beyond dinosaur extinction. This phenomenon is called nocturnal bottleneck [107] and affects not only CRY/PLs but also other light-associated proteins (such as opsins) and eye structures [108,109]. As a result, most mammals, including their famous representatives mouse and human, possess only MCRY (albeit duplicated as MCRY1 and MCRY2). Moreover, the presence of other DNA repair mechanisms, such as nucleotide excision repair (NER), appears to have allowed mammalian survival even in the absence of photolyases [110,111]. Interestingly, there is also a link between NER and MCRY as NER activity is regulated by the circadian clock and thus by MCRY [112,113,114]. However, most mammals belonging to Prototheria (for example the platypus *Ornithorhynchus anatinus*) as well as mammals belonging to Metatheria (as example the koala *Phascolarctos cinereus* or the common wombat *Vombatus ursinus*) still possess the CPDII photolyase in addition to the two MCRYs. In the long-nosed kangaroo *Potorous tridactylus*, we even found only one CPDII photolyase and no MCRYs. Interestingly, this CPDII is among the first photolyases discovered [115,116]. Experiments already showed that this one is an all-rounder that can not only repair DNA but also replace MCRY in MCRY-deficient mice [117,118]. Although we cannot exclude the possibility that *Potorous* indeed lacks MCRY due to the presence of this certain CPDII, we rather assume that MCRY1 and MCRY2 are lacking in *Potorous* because its genome has not yet been fully annotated. The CPDII PL sequence we obtained was derived from the photolyase-focused study of Yasui et al. (1994) [118].

In contrast, the Tasmanian devil *Sarcophilus harrisii* seems to lack the CPDII photolyase. Since the genome from the Tasmanian devil is fully sequenced and annotated, the absence of CPDII seems to be genuine. Thus, the Tasmanian devil is very similar to placental mammals (Eutheria) possessing only MCRY. Now, the question arises why most marsupials, also small and nocturnal in the Cretaceous, and in some cases still are, retained CPDII photolyase. Their geographic location might provide an explanation as most marsupials now live in Australia and in South America, which were quite close to the South Pole when Eutheria and Metatheria diverged about 120–140 million years ago [119,120,121]. Consequently, this resulted in stronger photoperiodic variation and seasonal effects due to the tilted earth axis [120]. Long summer days and the increased UV radiation may have contributed to the preservation of CPDII photolyase. In addition, lower temperatures at certain times in these areas may also have partly prevented the large poikilothermic dinosaurs from populating such habitats that were less problematic for the small homoiothermic mammals [120]. Thus, the latter may have remained diurnal.

Since CPDII has not been detected so far in Eutheria, it is assumed that its loss also occurred at the time when Eutheria and Metatheria separated [108]. The sperm whale *P. catodon* as a relatively recent mammal, however, challenges this hypothesis as it still possesses a CPDII photolyase (Appendix A). The split between the sperm whale and other terrestrial animals such as the hippopotamus possibly occurred only about 50 million years ago [122]. This would imply CPDII photolyases were conserved in Eutheria much longer than believed so far. However, none of the other animals studied in the Suborder Cetacea had CPDII. We even searched for possible CPDII photolyase homologs in all mammals already sequenced and annotated but were excluded from this study due to our limitations. Still, the sperm whale was the only organism that had a CPDII. We then reciprocally BLASTed the CPDII sequence of the sperm whale to find its closest relative. It was striking that, apart from a bird *Colinus virginianus*, the closest hits were parasites such as *Besnoitia besnoiti*, *Toxoplasma gondii*, *Cystoisospora suis* and *Neospora caninum*. This rather suggests that the whale tissue used for sequencing was likely contaminated with a parasite. This might also be true for the bird *Colinus virginianus*. Similarly, in our phylogenetic tree, the CPDII from the sperm whale clusters with the tapeworm *Rodentolepis nana* and not with other chordates. Furthermore, it should be emphasized that in contrast to the MCRY sequences, the CPDII sequence is a partial mRNA sequence and could not be assigned to any chromosome. Taken together, we strongly believe that CPDII in the sperm whale is an impurity and that none of the eutherians studied to date possess a CPDII photolyase. Thus, our study provides further confirmation of the bottleneck theory.

#### 4.3.6. The Arrow Worm Chaetognata Only Feature DCRY

The transparent arrow worm *S. cephaloptera* is the only sequenced species we have found in the phylum Chaetognata. It is a marine carnivore that belongs to the abundant planktonic organisms. Whether the Chaetognata belong to the Protostomia or whether they form a sister group to the Protostomia and Deuterostomia is still controversial [123,124]. According to paleontological studies, Chaetognatha already existed in the Early Cambrian (540-520 Ma ago) [125]. Thus, the Chaetognata might be among the first animals to possess a DCRY. Notably, we could not identify all other CRY/PL members in it. However, the DCRY of *S. cephaloptera* was identified in a study that focused on photoreceptors in the eyes [126]. Since the genome of this species has not yet been fully sequenced and annotated, it cannot be excluded that the MCRY and photolyases merely have not yet been identified. 

#### 4.3.7. Enriched CRY/PLs in Mollusks and in Annelids

Mollusks and annelids are representatives of the Protostomia Superhylum Spiralia. Both have an increased number of CRY/PL family members. Most of the mollusks and annelids possess five members of the CRY/PL family. These are MCRY, DCRY, DASH-CRY and the photolyases 6-4 and CPDII (Figure 4 and Figure 7). However, we did not find CPDII photolyases in the three sequenced marine mollusks belonging to the order Nudibranchia and no 6-4 photolyase in the East Asian common octopus *Octopus sinensis* (Appendix A). Some mollusks also lack the 6-4 photolyase or the DASH-CRY but MCRY and DCRY are very common in them. In most cases, the absence of the one or the other CRY/PL members is likely due to incomplete sequencing/annotation. Some mollusks even possess more than five CRY/PL members. While the freshwater snail *Biomphalaria glabrata*, for example possesses an additional CPDIII photolyase, six other mollusks including the Pacific oyster *Crassostrea gigas* possess an additional PCRY-like. In the case of *B. glabrata* the additional CPDIII photolyase, which occur very rare in metazoans, might stem from bacterial infections carried by its parasite *Schistosoma mansoni*, a platyhelminth [127,128]. The *Crassostrea* genome, on the other hand, is known to be highly polymorphic with transposable elements still actively shaping it [10]. This, coupled with the fact that all PCRY-like containing mollusks are marine animals being in constant contact with other microorganisms, makes the inclusion of PCRY-like via horizontal gene transfer more likely. We also find PCRY-like in the marine annelid Dumeril’s clam worm *Platynereis dumerilii*. The gain of PCRY-like again indicate a possible horizontal gene transfer. Besides their circadian clock, *P. dumerilii* is famous for its lunar clock, which characterizes their monthly reproductive cycles [129]. Whether the additional CRY/PL family members (besides the circadian MCRY and DCRY) play a role here is still elusive. However, the other two annelids lack PCRY-like. In the segmented worm *Capitella teleta*, only the MCRY, DCRY and the CPDII photolyase appear to be present. Since *C. teleta* has a well-sequenced genome [130], we exclude any sequencing errors and assume that the CRY/PL members are indeed missing. 

#### 4.3.8. Strong Reduction of CRY/PLs in Platyhelminths & Nematodes

The major CRY/PL member in platyhelminths and nematodes is the CPDII photolyase. All examined plathelminth species possess CPDII, and some additionally possess the 6-4 photolyase. The whipworms *Trichuris trichiura* and *Diploscapter pachys* are the only nematodes having a CPDI photolyase instead of CPDII. The typical animal CRYs MCRY and DCRY involved in the circadian clock are absent in platyhelminths and in nematodes studied. Still, they might have an intact clock. Circadian clocks were described in some platyhelminths by behavioral studies [131]. Since the organisms studied by them are not yet fully sequenced and annotated, their CRYs may still be undiscovered. However, incomplete sequencing and annotation can be ruled out for the best-known model organism *Caenorhabditis elegans*. Although *C. elegans* lacks all types of CRY/PL members, it still retains a functioning circadian clock [132,133]. Redox clocks based on S-sulfinylation of the peroxiredoxin (PRDX) protein represent potential alternatives for CRY-independent molecular clocks [134]. As nematodes and flatworms often live in darker, light-protected areas (such as the seafloor, in the soil, or as parasites inside a host), they are not highly exposed to light and thus might have reduced or even lost their CRY/PLs.

#### 4.3.9. Arthropoda: Varying Distribution of CRY/PL Members within the Chelicerata

The phylum of arthropods, consisting of Crustacea, Chelicerata, and Hexapoda, is one of the largest phyla studied and, accordingly, exhibits the greatest CRY/PL variability among species (Appendix A). It is questionable whether we would detect such differences in other phyla, with even similar numbers of animals. Finally, arthropods are not only the most species-rich phylum, but also include animals with diverse habitats and lifestyles. The ancient horseshoe crab *L. polyphemus* and the many spiders (order: Araneae) still possess MCRY, DCRY, and photolyases 6-4 and CDPII, which are very frequent and typical for arthropods. But the Arizona bark scorpion *Centruroides sculpturatus* (order: Scorpiones) and all ticks tested (subclass: Acari, order: Ixodida) lack nearly all CRY/PLs except MCRY. CPDI and CPDIII photolyases are also common in Chelicerata. While all studied mites from the order Sarcoptiformes possess only the CPDI photolyase, the sea spider *N. striatum* possesses a CPDIII photolyase. Moreover, all chelicerates tested lack DASH-CRY. 

In general, we find the greatest differences in the subclass of mites Acari. The anatomical, nutritional, and behavioral differences in them are reflected in their differently reduced CRY repertoire. While all ticks have only MCRY and Mesostigmata only CPDII photolyase, Sarcoptiformes have only CPDI photolyase. Since several animals within the orders possess their characteristic CRYs, we cannot assume any sequencing or annotation errors here. The three Trombidiformes examined are the only mites that differ from each other even within their order. The fact that *L. polyphemus* has four CRY/PLs suggests that the original Chelicerata also possessed multiple CRY/PLs. The large variation and reduction within the subphylum suggest specialization to specific CRY/PLs quite early in evolution (Appendix A, Figure 6). The hardened exoskeleton of some chelicerae could protect against UV radiation and thus allow a CRY/PL reduction. However, since crustaceans with hardened exoskeletons (e.g., the American lobster Homarus americanus) continue to have multiple CRY/PLs, this may not be the only reason. The light-protected way of life of some animals in dark places or even parasitic in a host could be another explanation. The parasitic lifestyle makes not only photolyases redundant, but also a separate circadian clock.

#### 4.3.10. DASH-CRY Is Quite Common in Crustacea

In contrast to chelicerates and hexapods, many crustaceans possess DASH-CRY in addition to MCRY, DCRY, and photolyases 6-4 and CPDII. These five are present in almost all Thecostraca, Branchiopoda, and Eucarida. Many crustaceans of the class Hexanauplia even have additional CRY/PLs that are rarely found in arthropods. While the aquatic copepod *Eurytemora affinis* has an additional PCRY-like, the salmon louse *Lepeophtheirus salmonis* possesses a CPDIII photolyase. At least for the aquatic copepod we believe a subsequent uptake via HGT. DCRY and DASH-CRY are completely absent in the six Peracarida. Many of them only possess MCRY, and sometimes CPDII photolyase too. Though this CRY/PL reduction could indicate a subsequent loss, we cannot exclude incomplete sequencing as a cause, too. We assume at least that the *H. azteca* data is complete, as this species has a BUSCO score of >95% in its annotation report. But we could only identify single mRNAs from clock-related studies for *Eurydice pulchra* and *Talitrus saltator* [135,136] and no whole genomes. In Eucarida, where we found all five CRY/PLs, we also find very good BUSCO scores. The only exception is the Antarctic krill *Euphausia superba*, in which we found only MCRY and DCRY. Since both sequences are from a clock-focused study [137], we believe that their photolyases are not sequenced and annotated yet (Appendix A). 

#### 4.3.11. MCRY, DCRY, 6-4 and CPDII Photolyases Are the Hexapod Typical CRY/PLs

The springtails *Orchesella cincta* and *Allacma fusca* are the only sequenced hexapods that belong to the class Collembola rather than Insecta. The fact that both possess a DASH-CRY suggests that the common ancestor of Crustacea and Hexapoda also possessed one. The silverleaf fly *Bemisia tabaci* (Hemiptera) and the fungus gnat *Bradysia coprophila* (Diptera), both pest species exposed to strong sunlight, still possess DASH-CRY. Since DASH-CRY is absent in all other insects studied, this must be a posterior acquisition, e.g., by horizontal gene transfer (HGT). HGT has already been described in several studies with *Bemisia* [138,139,140]. MCRY, DCRY, and photolyases 6-4 and CPDII are the most abundant CRY/PLs in insects, although they do not always occur together. While some orders possess all four, a reduction is observed in others. Such a reduction may be due to the fact that many published CRY/PL sequences are from CRY-focused studies (e.g., [141,142]) and thus at least some CRY/PLs may not have been discovered. This is supported by the fact that Kotwica-Rolinska et al. (2021), recently reported that the two-spotted cricket *Gryllus bimaculatus*, in which we found only MCRY and DCRY, also has both photolyases [43]. However, for insects that are generally nocturnal and hide in protected, dark places during the day (termites and cockroaches) or live parasitically hidden and protected (Phthiraptera), we assume that this loss is accurate. 

The CRY/PL distribution already differs within insects of individual orders, as exemplified by the Hemiptera. While the Sternorrhyncha (including aphids and whiteflies) have at least four CRY/PL, *Laodelphax striatella* and *Apolygus lucorum* lack 6-4 PL, and the fire bug *Pyrrhocoris apterus* additionally lacks 6-4 PL and DCRY. Similar reduction is seen in the Hymenoptera, which always lack DCRY and 6-4 PL, and sometime CPDII PL. The situation is similar in Coleoptera (Appendix A), the only exception being the jewel beetle *Agrilus planipennis*, which has all four insect CRY/PLs. This could be due to the lifestyle of the invasive beetle, which lays its rather unpigmented eggs in bark cracks of ash trees on the sunny side, where they need to be protected from sun damage. DCRY is the best known CRY of the Diptera. However, our analyses show that it is absent in the fungus gnat *B. coprophila*, which is curious given the recovery of DASH-CRY mentioned above. Perhaps the absence of DCRY favored the presence of DASH-CRY or vice versa. Another exception is the black soldier fly *Hermetia illucens*, which has photolyases 6-4 and CPDII (the typical CRY/PLs of dipterans) and MCRY in addition to DCRY. It is also the only Brachycera in our study that does not belong to the suborder Muscomorpha but to Stratiomyomorpha. Recently, other flies belonging to the suborder Muscomorpha were also found to possess MCRY [43]. Since the genomes of these flies have not yet been annotated, they are missing from our study. This finding clearly indicates that MCRY is lost only in certain superfamilies of Muscomorpha.

### 4.4. The Sudden Appearance and Disappearance of CRY/PL Members within Single Taxa

#### 4.4.1. Gene Duplication & Gene Loss

Gene duplication is an effective method to obtain genes with new functions without losing the original’s function. Thus, incomplete duplication of plant DASH-CRY likely resulted in the creation of an additional plant photolyase (PPL) without losing DASH-CRY. Gene duplicates with specialized functions are also known and already discussed for PCRY in land plants [143] and for MCRY in mammals [144,145]. Even MCRY itself certainly arose from gene duplications of 6-4 photolyases [38,66]. Our study suggests that ACRY, which occurs in anthozoans and poriferan-specific cryptochromes, has also arisen from incorrect 6-4 duplications and subsequent progression. The same is true for DCRY. But whether DCRY arose from a duplication of 6-4 photolyase or from MCRY we are unable to say at this time.

The emergence of various subfamilies is followed by specialization and the loss of individual subfamilies. The absence of various CRY/PL members during evolution is probably caused by gene loss. Single point mutations that affected the functionality of CRY/PLs may have ultimately led to complete loss of the gene. This was likely corroborated by additional factors such as altered living conditions (parasitic lifestyle, nocturnal activity, motility) and gain/loss of other genes (e.g., other repair mechanisms). As an example, many parasitic animals as well as those that live very closely with humans (like the body louse or bedbug), have generally fewer CRY/PLs than their relatives have. One reason for this could be that the clothing and dwellings that provide protection for humans also protect these animals. In addition, the nocturnal activity of small mammals in the mesozoic era, led to the loss of several CRY/PL members. Light activated enzymes have become expendable, as these animals were not exposed to light. Repairing UV-induced DNA damage may be one strategy, avoiding UV light another. 

#### 4.4.2. Horizontal Gene Transfer (HGT) & Microbiome

One reason for the sudden appearance of several CRY/PL family members in certain species may result from the events of horizontal gene transfer (HGT). As an example, the sea anemone *Nematostella vectensis* contains a CPDI sequence that is not present in the other Anthozoa and is very common in microorganisms. The transfer of certain bacterial genes into this anemone has been described previously [146,147]. 

Moreover, a recent study attempted to trace HGT events particularly in the CRY/PL family by examining inconsistencies in the phylogenetic tree [20]. By identifying a functional Trp tetrad in *Gloeobacterium*, the study suggested that this cyanobacterium might have taken up a 6-4 photolyase precursor from green algae. Indeed, *Gloeobacterium kilauensis*, included in our analysis, is one of the few bacterial species that has such a 6-4 homolog, further supporting this hypothesis. This kind of gene transfer ensures perfect adaptation to host species, which is especially crucial for symbiotic or parasymbiotic organisms. The DASH-CRYs of the whitefly *B. tabaci* or the fungus gnat *Bradysia*, both plant pests, could also indicate such an adaptation. That the uptake of a plant gene by the whitefly allows its detoxification from plant defenses has been already shown [148].

PCRY-like is most remarkable with respect to HGT. Its presence does not depend on the phylogeny, but rather on the living environment of the organisms. All organisms with PCRY-like live mainly in aqueous environments. First, we find PCRY-like in various algae, probably this is also their place of origin. Then we find PCRY-like sporadically in other marine organisms such as in the pacific oyster *C. gigas*, in the annelid *P. dumerilii* as well as in the marine copepod *E. affinis*. We also find PCRY-like in vertebrates, in this case mainly in fishes belonging to the class *Actinopteri* and in amphibians. Since no phylogenetic lineage between organisms is apparent, we assume that the more complex organisms possessing PCRY-like have acquired them subsequently by horizontal gene transfer. Furthermore, aquatic habitats have already been shown to promote HGT [56,57,149]. Remarkably, nearly all animals examined that have a PCRY-like also possess a DASH-CRY (Appendix A). However, the interplay between the two cryptochromes has not yet been explored.

Perhaps the inclusion of certain genes in one’s own genome is not even necessary if their function can be outsourced to the microbiome. This might not be possible for the DNA-repairing photolyases, but it would be possible for the cryptochromes involved in the circadian clock. After all, microbiomes can interact with their host by means of their metabolites [150,151] and thus also influence their host’s rhythmicity. Indeed, several studies suggest that a proper clock of the microbiome could complement or even replace that of the host organism [152,153,154,155].

#### 4.4.3. Do Ancient Animals Have More CRY/PL Family Members Than the Recent Ones?

Nowadays, organisms with few and with many CRY/PL family members coexist. Our results suggest that the more primitive animals (as example springtails) tend to have more CRY/PLs than their recent relatives (like flies, bees, or ants). Because organisms are constantly evolving, species with few or more CRY/PLs may have coexisted in earlier periods of evolutionary history as well. Morphological and habitat changes could have occurred that reduced the need for CRY/PLs. But why do most ancestral species present today tend to have more CRY/PLs? One reason is certainly the increased UV radiation at the beginning of life (3500 Mya), which was about 500 times higher than today and consequently caused DNA damages making photolyases indispensable [47,156]. A further reason includes the severe collapses occurring later in Earth’s history resulted in mass extinctions. At least the extinction events 250 Ma and 359 Ma ago were associated with the destruction of the ozone layer and a consequently increased UV radiation [157,158,159]. The destruction of the ozone layer may also have contributed to the extinction of the dinosaurs 66 million years ago, albeit due to an asteroid impact and associated air pollution. Under such conditions, organisms with more CRY/PL members, especially photolyases, certainly have a survival advantage, whereas for organisms without photolyases this could be fatal. An increased number of DNA mutations in organisms around these events has been described previously [159]. Organisms with fewer CRY/PLs may have died out due to increased DNA damage. Thus, we suspect that daylight-exposed species that survived these mass extinctions have relatively high numbers of CRY/PLs. This is certainly true for plants being highly dependent on sunlight.

#### 4.4.4. Methodical Errors: Incomplete Annotations & Possible DNA Contamination

For completeness, we must also mention that the sudden appearance and disappearance of several genes may also be due to methodological errors. Not all species in this study are fully sequenced and annotated. Some of the annotated sequences we used were obtained from clock-related studies in which only cryptochromes were examined and sequenced. Therefore, we cannot exclude the presence of photolyases in these species that have not yet been fully sequenced and annotated. Additionally, DNA contamination due to microbial infection could also be responsible for the sudden appearance of certain genes. CPDI and CPDIII photolyase are rarely found in metazoans but are very abundant in bacteria and fungi. In some metazoans, we find sporadic CPDI and CPDIII occurrences that often do not result in an obvious pattern. Another example of such contamination is the sperm whale *P. catodon*, which is probably infested with a parasite that may have had a CPDII photolyase (see 4.3.5). 

### 4.5. When and Why Did the Repair Enzyme Turn into a Clock Protein?

Except for CRY4, which seems to be involved in magnetoreception, all other CRYs appear to be involved in the circadian clock. CRYs have two unique properties that make them ideal for the circadian clock. First, as close relatives of photolyases, they have DNA-binding properties allowing them to enter the nucleus and influence the transcription of various genes. Second, they are photosensitive, allowing CRY-expressing cells to sense their environment and adjust their circadian clocks accordingly. The second ability is particularly important for multicellular organisms as only few of their cells are in direct contact with their environment. Photolyases already possess the two functions. It seems that different CRYs have consistently acquired their function convergently in the circadian clock. Plant CRYs originating from CPDIII PL, as well as animal CRYs originating from 6-4 PL are involved in the circadian clock. Also, the DASH-CRY in the fungus *N. crassa* seems to be involved in circadian clocks [27]. It is questionable whether one can already speak of a kind of clock for the photolyases of prokaryotes. Clearly, the enzymes activated by blue light perceive their environment in their own way and pass this information on to the cell.

The best known MCRY and DCRY, however, seem to have chosen only one of the properties of photolyases. While MCRY retained DNA binding ability and lost photosensitivity, DCRY adapted the opposite path. The question here is which of the functions is ancestral and characteristic of the circadian clock. Yuan et al. (2007), proposed that photoreceptor-like DCRY was the first animal CRY and that gene duplication gave rise to the DNA-binding MCRY form [66]. This contrasts with our results in which we have already found MCRYs in cnidarians, protostomes, and deuterostomes, i.e., all planulozoans, so that we assume that MCRYs is the more ancestral variant. If we look at its functionality, we find that DCRY has light sensitivity in common with 6-4 photolyase, but its function in the circadian clock in common with MCRY. However, whether DCRYs are derived from MCRYs by duplication or are also from the 6-4 PL is a difficult question.

Sequence analysis of the brachiopod *L. anatina*, in which we were able to identify MCRY and DCRY simultaneously for the first time, showed similar amino acid identities between DCRY and MCRY as well as between DCRY and 6-4 photolyases. In mollusks, DCRY seems to be more similar to MCRY than to 6-4 photolyases, while in insects the reverse is true and the DCRY is more similar to 6-4 photolyases. In the phylogenetic tree, DCRY forms a precisely separated cluster, but similarly far from MCRY and 6-4 photolyase. If we look at its functionality, we find that DCRY has light sensitivity in common with 6-4 photolyase, but its function in the circadian clock in common with MCRY. Now there are two possibilities. The first one is that DCRY originated from 6-4 PL and later found its role in the circadian clock and the second one is that it originated from MCRY and regained its light sensitivity. In this context, we also must mention that it is currently not yet clear whether all MCRYs are light-insensitive. At least a recent study indicates a possible light sensitivity of the mammalian MCRY2 [160]. Therefore, the possibility that DCRY is derived from a MCRY that was still light-sensitive but already active in the circadian clock cannot be ruled out.

Furthermore, we found only the DCRY and no MCRY in several organisms. In most species, this could be due to sequencing and annotation errors. Our study and the study by Kotwica-Rolinska et al. [43], clearly demonstrate that some flies still possess an MCRY. As yet, we confidently confirm that many fly species, including *D. melanogaster*, possess only the DCRY and no MCRY. Since they evolved relatively late in evolutionary history and some of them still possess MCRY, we assume that the loss of MCRY also occurred relatively late. However, because the DCRY-based PERIOD-TIMELESS clock appeared in one of the best-studied model organisms, *D. melanogaster*, it has naturally been well examined [161]. The MCRY-based PERIOD-MCRY clock and other combinations of MCRY- and DCRY-based clocks seem to be rather the rule. However, at least in cnidarians lacking both TIM and PER, also clocks consisting only of MCRY or MCRY-ACRY are conceivable [80]. Even in the more primitive sponges appearing also to lack PER and TIM, species-specific CRYs continue to play a role in the circadian clock [76,77]. CRYs appear to be more primordially involved in the circadian clock than their common associates PERIOD, TIMELESS, CLOCK, and CYCLE. 

Although there are exceptions such as the nematode *C. elegans*, which seems to have circadian clocks without cryptochromes, for most organisms the members of the CRY/PL family seem to form a solid basis for molecular clocks. Not only in the animal kingdom, but in all living beings, independent new clocks seem to have emerged again and again, very often starting from the CRY/PL family members. That the evolution of cryptochromes from photolyases is not yet complete is shown by the marsupial *P. tridactylus*, whose CPDII photolyase has also been shown to play a role in its circadian clock [117]. Taken together, although the famous *Drosophila* and the mammalian circadian systems are the best-studied clocks, they are far from being the only ones. The modular kits provided by the CRY/PL family, most notably, offer several ways to construct a molecular clock.

## 5. Conclusions

Our study reveals that members of the CRY/PL family are important, but not essential for life. Indeed, not all organisms have all CRY/PL members; even complete loss seems possible. After all, there are other ways to protect against UV damage and to establish circadian rhythms. UV damage can be prevented by avoiding sunlight and can be repaired with other repair systems as well [162] and circadian clocks can also be achieved by redox clocks and constituents of a partnering microbiome for instance [152,163,164]. However, CRY/PLs appears to offer increased potential for evolutionary survival, as many ancestral species living today have more CRY/PLs than their modern relatives. Furthermore, our study clearly reveals that various cryptochromes evolved independently from different photolyases, explaining their different ways of modes. Two properties of photolyases, (a) the perception of light and (b) the interaction with DNA, are crucial for the evolution of cryptochromes. The well-studied modern clocks in mice and *Drosophila* are far from standard and represent only a fraction of the complex ways to build a molecular clock. Interestingly, almost all cryptochromes influence the circadian clock. Which connection do photolyases and cryptochromes have? While photolyases repair UV-induced damage after the fact, cryptochromes can protect organisms in advance via the circadian clock. Nocturnal activity, avoiding the midday sun, and circadian expression of several UV-protective, repair, and control proteins supports and even relieves photolyase function, explaining the independent emergence of various cryptochromes from different photolyases.

## Figures and Tables

**Figure 1 genes-13-01613-f001:**
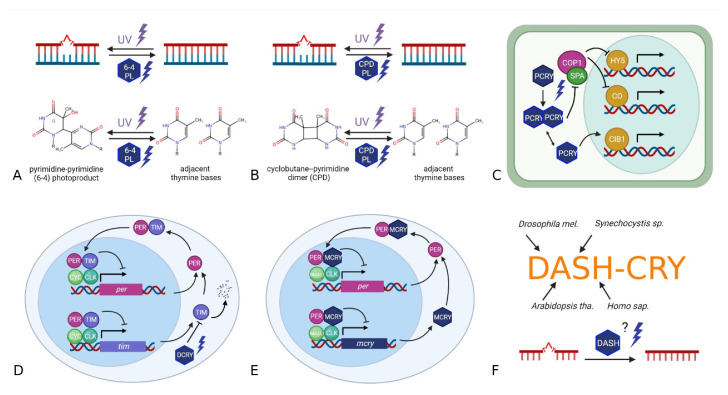
Schematic overview of six main members of the CRY/PL family based on their function in major model organisms and references given in the introduction. (**A**,**B**) The photosensitive photolyases repair UV-induced pyrimidine-pyrimidone dimer mutations in DNA using blue light. Depending on the type of mutation they repair, they are called 6-4 photolyases (pyrimidine-pyrimidone-6-4 photoproduct) or CPD photolyases (cyclobutane-pyrimidine dimer). Plant PCRYs (**C**) are photosensitive and directly or indirectly affect the transcription of several genes whose functions are also linked to the circadian clock. The Drosophila light-sensitive cryptochrome (DCRY) is activated by blue light and destabilizes TIMELESS, resulting in a resetting of the circadian clock (**D**). The mammalian light insensitive MCRY, which is also involved in the circadian clock, acts as an indirect transcription factor, and inhibits its own expression (**E**). DASH-CRYs (**F**) act as a link between different members of the CRY/PL family. They appear to be involved in both the circadian clock and the repair of mutations in single-stranded DNA.

**Figure 2 genes-13-01613-f002:**
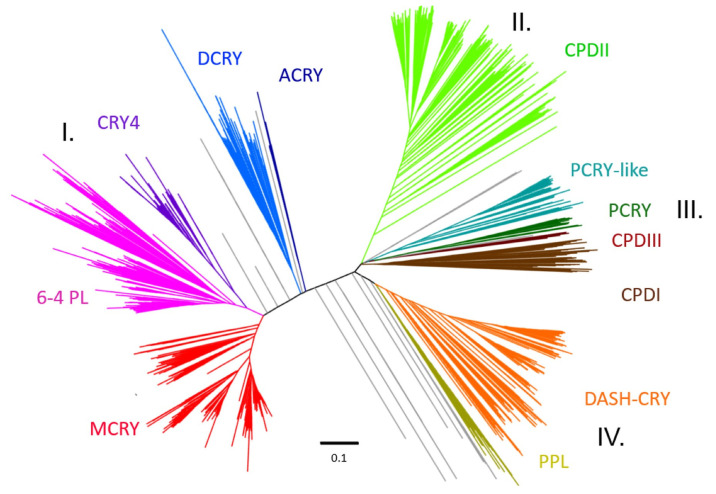
Unrooted phylogenetic tree revealing the four main clusters formed by CRY/PL family members. The 6-4 photolyase cluster (I) includes the mammalian type MCRY (red), the Drosophila-type DCRY (blue), CRY4 involved in magnetoreception (purple) and the 6-4 photolyase itself (pink). The CPDII cluster (II) consists of CPDII photolyases (light green). The CPDI/III cluster (III) includes the CPDI (dark brown) & CPDIII photolyase (light brown), the plant cryptochrome PCRY (dark green), and the PCRY-like (turquoise). The DASH cluster (IV) consists of DASH-CRY (orange) and the plant photolyase PPL (yellow). Branches we could not precisely assign are shown in grey.

**Figure 3 genes-13-01613-f003:**
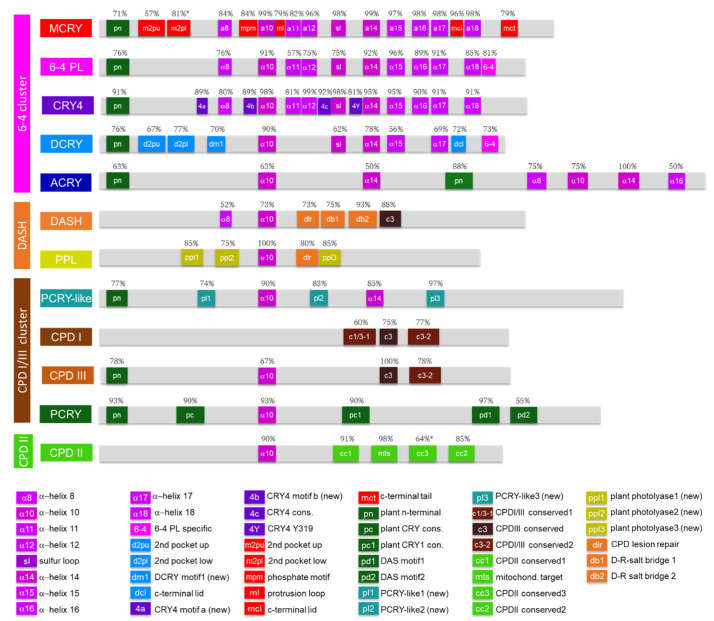
Schematic representation of the CRY/PL subfamilies along with their most frequent motifs. Only motifs that occur with a frequency of at least 50% are shown. A detailed list of all motifs, their sequences and references can be found in Appendix A, and protein sequence alignments of all subfamilies can be found in Appendix A. In the case of m2pl*, both insect (13%) and non-insect (68%) data were combined, and for cc3*, data from microorganisms & animals (59%) were summed with those from plants (5%).

**Figure 4 genes-13-01613-f004:**
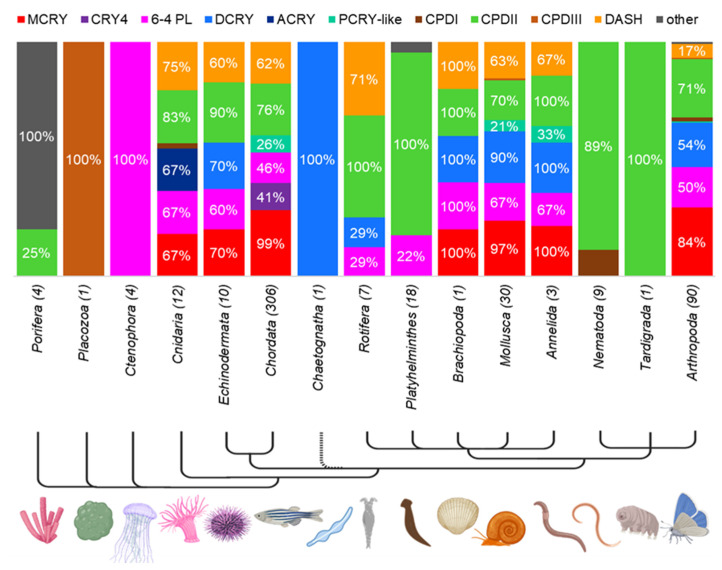
CRY/PL distribution within the metazoan phylum. The prevalence of a subfamily within a phylum is indicated in percent, with only percentages > 10% shown for clarity. The total number of animals examined in each phylum is given in parentheses beside the phylum name. See Appendix A for more details.

**Figure 5 genes-13-01613-f005:**
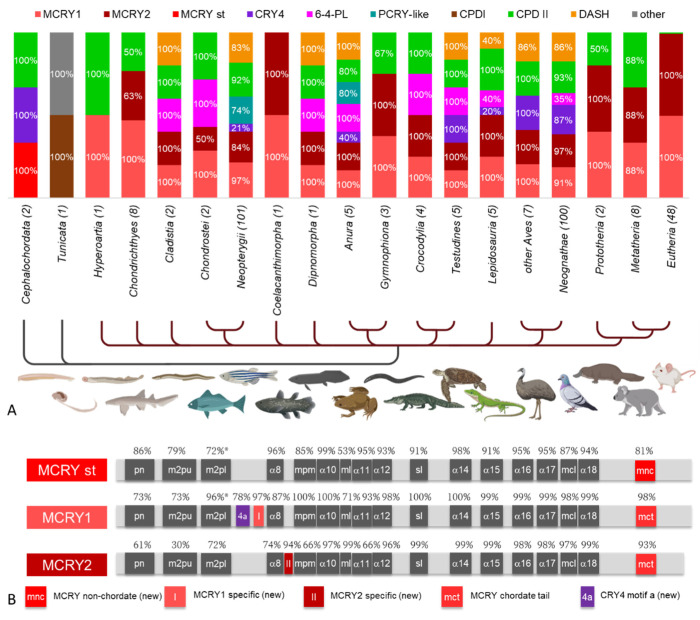
(**A**) Distribution of CRY/PL subfamilies within Chordata. Cephalochordata and Tunicata are shown with gray lines, while animals belonging to the subphylum Craniata are labeled with dark red lines. The prevalence of a CRY/PL subfamily within a taxon is given as a percentage. For clarity, only percentages above 10% are given. The total number of animals studied in each taxon is given in parentheses next to its name. Nearly all animals studied, with only a few exceptions, have an MCRY duplication leading to MCRY1 (light red) and MCRY2 (dark red). (**B**) Characteristic motifs of MCRY1, MCRY2 and the standard MCRYs (MCRY st) found in non-chordates. Type-specific motifs are shown in color. The motifs mnc, I, II, and 4a are novel motifs. All other motifs are explained in Figure 3.

**Figure 6 genes-13-01613-f006:**
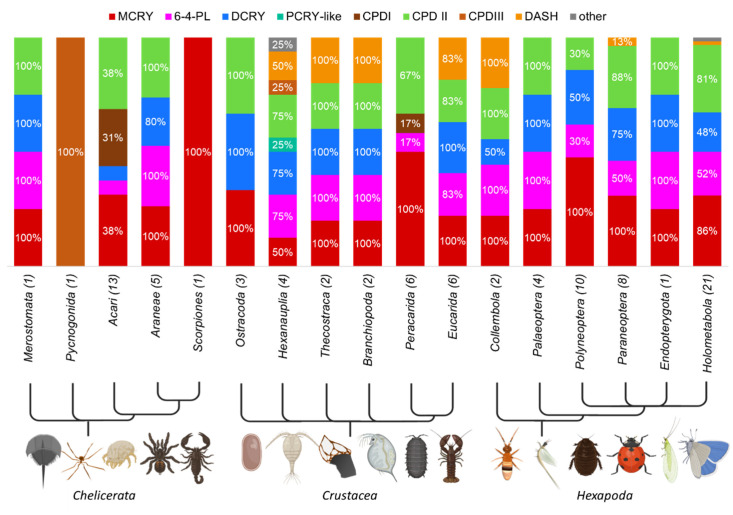
Distribution of CRY/PL subfamilies within the orders of Arthropoda. The occurrence rate of a subfamily within an order is given as a percentage, only percentages > 10% are shown for clarity. The total number of animals studied in each phylum is given in parentheses next to the order name. Chelicerates, crustaceans, and hexapods form the main arthropod subgroup. The major CRY/PL members of arthropods are MCRY, DCRY, and the photolyases 6-4 and CPDII. Many crustaceans also carry the DASH CRY. The four major CRY/PLs are found in all subgroups of the Hexapoda substrain, even though they are not equally abundant in all groups. MCRY is the most important CRY/PL of the hexapods. In Collembola and in individual insects of the cohort Paraneoptera and Holometabola the DASH-CRY is also present.

**Figure 7 genes-13-01613-f007:**
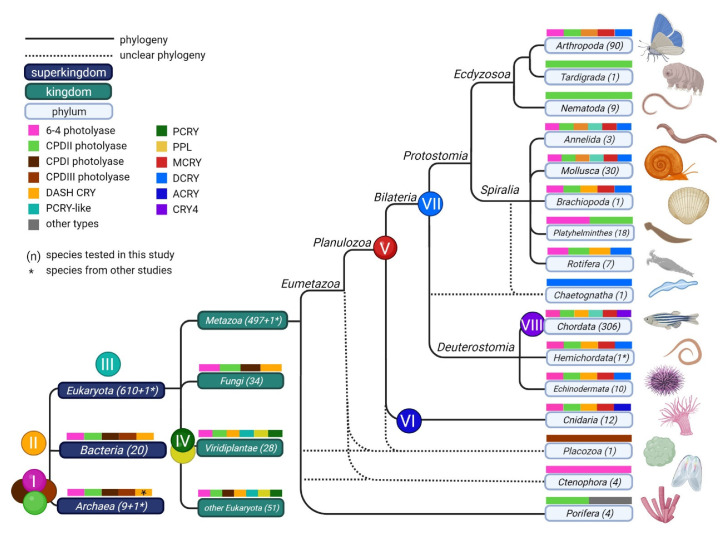
The origin of the CRY/PL members. (I) All three photolyases 6-4, CPDII and CPDI/III are found in all three domains of Eukaryota, Bacteria and Archaea. (II) DASH-CRY is mainly found in Eukaryota and in Bacteria. However, one archaeon is known to have DASH-CRY as well. (III) In eukaryotes, the PCRY-like protein arose next. It is found in all kingdoms except fungi. We assume that it has been lost in the fungi studied. (IV) Origin of PCRY and PPL in some algae and in Viridiplantae. (V) Origin of MCRY, which occurs in both Cnidaria and Bilateria. Its origin is therefore attributed to planulozoans. (VI) Origin of ACRY in certain cnidarians. (VII) Origin of DCRY, which occurs only in Bilateria, but in both Deuterostomia and Protostomia. (VIII) Origin of CRY4 in some chordates. The number of organisms tested is shown in parentheses next to the taxonomy name. The archaeon *Halorhabdus utahensis* and the hemichordate *Ptychodera flava* (both marked with *) were investigated in the studies by Mei and Dvornyk (2015) [19] and Kotwica-Rolinska et al. (2021) [43].

**Table 1 genes-13-01613-t001:** Name, accession number, the CRY/PL type, and the species origin of the proteins used as reference for our phylogenetic tree analysis are listed.

**CRY/PL Type**	**Species**	**Name**	**Accession No.**
DCRY	*Drosophila melanogaster*	CRY	NP_732407.1
6-4 PL	*Drosophila melanogaster*	6-4 PHR	BAA12067.1
MCRY	*Mus musculus*	CRY1	NP_031797.1
DASH	*Synechocystis* sp. *PCC 6803*	DASH	AGF51454.1
CPDI	*Escherichia coli*	PHOTOLYASE	WP_001583322.1
CPDII	*Drosophila melanogaster*	PHR	NP_523653.2
CPDIII	*Agrobacterium fabrum*	PHOTOLYASE	WP_174020122.1
PCRY	*Arabidopsis thaliana*	CRY2	AAL16379.1
PCRY-like	*Phaeodactylum tricornutum*	CryP	XP_002179379.1
PPL	*Arabidopsis thaliana*	PHR2	NP_182281.1
CRY4	*Danio rerio*	CRY4	XP_005168334.1

**Table 2 genes-13-01613-t002:** The frequency with which a member of the CRY/PL family occurs in the individual kingdoms is shown. The three domains Archaea, Bacteria and Eukaryota are listed, whereby the domain Eukaryota has been further subdivided into its kingdoms Metazoa, Fungi, Viridiplantae and other unicellular Eukaryota. The number of organisms tested per kingdom is given in brackets next to the taxon name.

		MCRY	CRY4	6-4-PL	DCRY	ACRY	PCRY like	CPDI	CPDII	CPDIII	PCRY	DASH	PPL	other
*Eukaryota*	*Metazoa (497)*	86%	25%	47%	18%	2%	18%	2%	76%	1%	-	50%	-	2%
	*Fungi* *(34)*	-	-	65%	-	-	-	85%	12%	-	-	53%	-	-
	*Viridiplantae* *(28)*	-	-	86%	-	-	11%	4%	86%	4%	75%	82%	61%	-
	*unicellular Eukaryota (51)*	-	-	63%	-	-	25%	14%	73%	2%	-	65%	6%	22%
*Bacteria (20)*		-	-	15%	-	-	-	65%	25%	15%	-	30%	-	-
*Archea (9)*		-	-	33%	-	-	-	22%	44%	11%	-	-	-	-

## Data Availability

All data presented in this study are available in the article itself or in the provided Appendix A.

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
