# Peer review of "The Gain and Loss of Cryptochrome/Photolyase Family Members during Evolution"

_genes, 2022, doi:10.3390/genes13091613_

Round 1

Reviewer 1 Report

This study extensively analyzed Cryptochrome/Photolyase family member proteins by using phylogenetic tree and 12 motif analyses. Although similar researches have been published, this study will provide useful imformation to researchers in circadian clock and DNA repair fields. I just recommend authors to include following paper in reference list. This paper is the first report of cryptochrome in animal.

Todo T, Ryo H, et al. Science. 1996 Apr 5;272(5258):109-12. doi: 10.1126/science.272.5258.109.PMID: 8600518

Similarity among the Drosophila (6-4)photolyase, a human photolyase homolog, and the DNA photolyase-blue-light photoreceptor family.

Reviewer 2 Report

This is a well-written, comprehensive contemporary analysis of the Cryptochrome/Photolyase family members among various metazoan lineages, discussing their potential evolutionary course. The article is clear, the information is presented skillfully, and the figures are good.

A few minor amendments are indicated:

lines 366-376, The sentence “ Only in the dimorphic fungus Mucor lusitanicus and in all four microsporidia examined we could not detect CPDI.” should be changed to “However, the dimorphic fungus Mucor lusitanicus appears to be the exception among the four microsporidia that were examined as CPDI could not be detected.”

line 381 – 382, correction of that reference is necessary.

line 408-409 – same

line 595-596 – same

line 673-674 – same

line 807-808 – same

line 444, change “..... possesses additionally a … “, to “..... possesses an additional CPDI........”

line 853 – place an 'a' between 'for' and 'very'

line 877, “Most birds studied have besides the two MCRYs, DASH-CRY also the CPDII photolyase” is awkward. Suggest, “Besides the two MCRYs, most birds also have DASH-CRY and CPDII photolyase.”

line 927, suggest change to “ Since CPDII has not been detected so far......” i.e. 'was' to 'has'

line 979, remove ). at the start of the sentence.

Line 1166 - 1167, place the word 'A' in front of 'Further' and change 'occurred' to 'occurring' and 'resulting' to 'that resulted'.

line 1266, suggest rephrasing to '….... redox clocks and constituents of a partnering microbiome for instance.” with a further reference, such as: Frazier, K., & Chang, E. B. (2020). Intersection of the gut microbiome and circadian rhythms in metabolism. Trends in Endocrinology & Metabolism, 31(1), 25-36.

The comments below do not require any current amendment or reply since it is probable that this type of analysis would be beyond the scope and tools of the investigators at this time. However, modern evolutionary analysis should prompt an integration of body cell circadian clock function as correlative with a sample constituent microbiome. Obviously, this is very difficult, but fuller answers lie through its inclusion. For example, some potential questions for a more in-depth analysis, building on this current research might include”

In discussing HGT and the acquisition of PCRY-like, what role might their constituent microbiomes have played in that transfer process?

Is it possible that metazoans have retained a reliance on more cytochromes than accounted for in their respective genomes by outsourcing some functions to their constituent microbiome in the same way that the production of mammalian serotonin relies on microbial partners within the gut microbiome, i.e. might some of the missing cytochrome genetic functions be supplied by obligate microbial constituents within the various organism's microbiomes?

Reviewer 3 Report

The Gain and Loss of Cryptochrome/Photolyase family members during Evolution

General remarks

The manuscript entitled " The Gain and Loss of Cryptochrome/Photolyase family members during Evolution” perform a bioinformatics search for genes of the Cryptochrome/Photolyase family

Abstract section

Comment 1. In line 12, What type of taxonomic groups and number of sequences per group were analyzed? What type of sequences were analyzed, and were they obtained from in-house sequencing or were they downloaded from databases?

Comment 2. In line 13, What type of bioinformatics methodology was used in the identification of these genes?

Comment 3. In line 17, What methodology was used in this assignation?

Introduction section

In general, the introduction is clear and well structured; however, some aspects should be expanded.

Comment 4. In line 33, Please indicate the reference

Comment 5. In line 38, Is this process conserved in all metazoan species?

Comment 6. In line 48, Again: Is this process conserved in all metazoan species? or, Is there any difference between the taxa? In which organisms do they fit the model presented in Figure 1A and 1B?  What do these changes imply?

Comment 7. In line 64, in Figure 1, Was figure 1 constructed by the authors based on previous results?  Which authors?  Is it based only on the Drosophila model?  reference. Are there other models that confirm this conserved process?

Comment 8. In lines 122 to 126, What is the gap in these studies and how do you justify this work?

Materials and methods: This section present several methodological aspects that need to be explained. many of these aspects are crucial to validate the results presented, so they must be sufficiently resolved.

For example, in this section, several aspects should be indicated: a) origin of the sequences (complete or partial genome, transcriptome, ect); b) number of sequences analyzed total and per taxonomic group (e.g. mammals, fish, birds, amphibians, arthropods, plants...etc)

Comment 9. In lines 133 to 134, the methodology in the identity of these genes must be sufficiently described. In this case, which was the criterion for inclusion and exclusion of sequences for this analysis?

Comment 10. In lines 140 to 141, why not BLASTP local?

Comment 11. In line 142, why not the whole gene family by reference species? Justify

Comment 12. In line 148, It is not clear how the authors manage to evade the blastp finding within such phylogenetically distant organisms. This is a central point to the proposed methodology and must be well supported.

Comment 13. In line 154, in Phylogenetic Tree section. Serious methodological gaps are detected in the phylogenetic reconstruction that must be sufficiently clarified.

Comment 14. In line 155, How the authors justify that the phylogeny was constructed with the geneious program. why not with IB or ML in other more robust programs?

Comment 15. In line 156, Why did you use this aligner and not another one like Muscle, Tcoffee or Clustal Omega alignment programs?

Comment 16. In line 164: where did the results come from?

Comment 17. In line 170, In Identification of cryptochrome/photolyase subfamilies section:  What cut-off points were taken in the blastp and other methodology to confirm presence and/or absence of genes and their copies? What bioinformatics tools were taken into account to avoid false positives or negatives of gene copies?

Comment 18. In line 177. In Characterization of CRY/PL family sequences via their protein motifs section: What was the motif sequence used in each taxonomic group? What level of sensitivity was used?

Results and Discussion section

although the results and discussion sections are well structured, presenting interesting results; however, these must be sufficiently validated from the methodology. therefore, each of the remarks in the methodology must be answered in order for these results to be supported.
